# Visuomotor decision-making through multifeature convergence in the larval zebrafish hindbrain

Katja Slangewal [1,2,3] ✉, Sophie Aimon [1,2], Maxim Q. Capelle [1,2,3], Florian Kämpf [1,7], Heike Naumann [1], Krasimir Slanchev [4], Herwig Baier [4] & Armin Bahl [1,2,3,5,6] ✉

Animals continuously extract and evaluate diverse sensory information from the environment to guide behavior. Yet, how neural circuits integrate multiple, potentially conflicting, inputs remains poorly understood. Here, we use larval zebrafish to address this question, leveraging their robust optomotor response to coherent random dot motion and phototaxis towards light. We demonstrate that animals employ an additive behavioral algorithm of three visual features: motion coherence, luminance level, and changes in luminance. Using brain-wide two-photon imaging, we identify the loci of these computations, with the anterior hindbrain emerging as a multifeature integration hub. Through single-cell neurotransmitter and morphological analyses of functionally identified neurons, we characterize potential connections within and across computational nodes. These experiments reveal three parallel and converging computational pathways, matching our behavioral results. Our study provides a mechanistic brain-wide account of how a vertebrate brain integrates multiple features to drive sensorimotor decisions, bridging behavioral algorithms with their neural implementation.

Animals continuously process noisy and often conflicting sensory information from their complex, dynamic environment to guide movement decisions. The precise neural mechanisms that allow the brain to extract and computationally process such cues remain poorly understood.

Multiple sensory streams can be combined through, for example, additive or winner-takes-all algorithms. For instance, when an animal attends to different targets, directing its gaze toward the averaged location can provide an effective solution[1,2]. Such an additive strategy has been observed across various species[3–7]. In experiments where animals need to combine multiple sensory features during a motor control decision-making task, additive mechanisms reliably encode signals and guide behavioral choices[8]. Alternatively, parallel representations of potential actions can compete for dominance[9], and, ultimately, a winner-takes-all mechanism can resolve conflicting information. This competition is also thought to be fundamental to stimulus-driven attention[10] and is widely implemented across the animal kingdom[11–15]. How these different computational strategies, adding information and winner-takes-all, are mechanistically represented across the brain and how they control behavior remains an open question.

Sensory input reaches the decision centers in the brain through different sensory modalities, like vision or olfaction, and through distinct stimulus features detected by the same sensor within one

[1]Department of Biology, University of Konstanz, Konstanz, Germany. [2]Centre for the Advanced Study of Collective Behaviour, Konstanz, Germany. [3]International Max Planck Research School for Quantitative Behaviour, Ecology and Evolution (IMPRS/QBEE); Max Planck Institute of Animal Behavior, Konstanz, Germany. [4]Max Planck Institute for Biological Intelligence, Martinsried, Germany. [5]Max Planck Institute of Animal Behavior, Konstanz, Germany. [6]Zukunftskolleg, Konstanz, Germany. [7]Present address: Medical Research Council Laboratory of Molecular Biology, Cambridge, UK. ✉e-mail: katja.slangewal@uni-konstanz.de; armin.bahl@uni-konstanz.de

modality. Higher-order neural activity then drives a specific motor action or an adjustment of ongoing behavior. A key question is whether stimuli that cause the same behavior converge at an early stage or remain represented in parallel throughout the processing hierarchy. Early convergence reduces energy demands and enhances computational efficiency, while parallel representation allows for state-dependent behavioral flexibility.

To address this topic, larval zebrafish, with their accessible nervous system, form an effective model to dissect the behavioral algorithms and neural mechanisms that underlie multifeature sensory processing within the visual modality. Previous studies in larval zebrafish have explored the integration of separate sensory stimuli, identifying the anterior hindbrain and adjacent regions as a key site of higher-order sensory processing and behavioral control: Temporal integration of visual motion cues[16,17], processing of heat[18], of futility[19], of luminance[20,21], of internal states related to the optokinetic reflex[22], as well as of opposing threatening looming cues[11]. In addition, this brain area controls exploratory locomotion states[23] and represents integrated heading direction information for behavioral navigation[24]. Using a variety of visual stimuli within a single experiment, it has also been shown that stimulus representations overlap within the anterior hindbrain region[20]. However, it remains unclear how this brain area contributes to the extraction of visual features when multiple stimulus types are presented simultaneously and how it resolves situations in which sensory cues are presented in a conflicting configuration.

In this work, we disentangle how motion and luminance are integrated simultaneously in the larval zebrafish brain, combining two ethologically relevant behaviors: the optomotor response and phototaxis. The optomotor response is a position-stabilizing behavior in response to whole-field visual motion in which fish prefer to follow the direction of motion[25]. Whereas phototaxis allows larvae to navigate to optimal light conditions, usually preferring brighter parts[26,27]. Both stimuli can drive directional swimming[26,28–31], and previous work has provided a good understanding of the brain regions and computations involved[16,17,20,25,32–35]. Our goal is to determine how these circuits interact. Are visual motion and luminance cues processed in parallel, and where do representations converge? Or are signals combined early on according to their directional influence on behavioral choice? Does a winner-takes-all competition resolve sensory conflict, or are inputs simply added to guide motor decisions? Are conflicting stimulus situations processed differently from congruent ones?

In this work, we address these questions by first using precisely controlled behavioral experiments to construct a multifeature integration model. We then use our model to generate specific circuit hypotheses about the underlying neural computations in the brain. Based on such information, through brain-wide functional microscopy, we then search for respective representations. Starting at the identified loci, we then characterize neuronal anatomies, neurotransmitter distributions, and potential connectivity arrangements across the computational units, allowing us to update and further refine our model. Thus, our study provides a comprehensive account that links behavioral experiments to neural circuit structure and function, enabling us to uncover a biologically plausible algorithmic and mechanistic model of multifeature sensorimotor decision-making in larval zebrafish.

## Results

### Addition of multifeature sensory cues captures sensorimotor decision-making

Larval zebrafish swim in discrete bouts and adjust their heading direction spontaneously and based on external sensory cues. While visual motion drift and spatial luminance cues are both known to directionally guide animals, it remains unclear how zebrafish integrate these two stimuli when presented at the same time. Do they prioritize the strongest cue through a winner-takes-all strategy, or do they sum inputs across the different sensory features? To investigate this question, we systematically paired coherent random dot motion stimuli of varying strengths with either a congruent or a conflicting spatial luminance cue.

We presented visual stimuli to the bottom of 12 cm diameter arenas where unconstrained individual larvae could freely swim. A high-speed camera tracked animals in real time (Fig. 1a and Supplementary Movie 1). We detected swim bouts based on increases in the variance of body orientation (Fig. 1b), with each bout considered a decision event. To simplify our analysis, we binarized bouts based on orientation change; left (<−2°) or right (>2°) (Fig. 1c, top). Swim bouts occurred in stochastic interbout intervals at around 2 Hz (Fig. 1c, bottom). To allow for variations of strength in the motion stimulus, we used random dot kinematograms of varying coherence levels.

Certain fractions of dots moved coherently right- or leftward. All dots had short lifetimes and got randomly repositioned in the arena, preventing animals from tracking individual dots. We superimposed coherent motion stimuli with lateral luminance cues and locked stimuli to the body orientation of animals in real-time (Fig. 1d). This design ensures that during the stimulus, input into the visual system remains stable over time from the perspective of the fish, providing a consistent and well-controlled configuration for the study of sensorimotor decision-making. We initially focused our analysis of directional swimming during the steady-state phase within the last 10 s of stimulus presentation.

To uncover the computational principles underlying multifeature stimulus integration, we analyzed zebrafish behavior as a function of coherent motion strength with or without superimposed lateralized luminance cues. With higher coherence lelvels, animals should be more likely to follow motion cues[16,17]. The behavioral decision curve in response to motion alone (= only-motion) resembles a psychometric curve (Fig. 1e,f) in which stronger rightward motion results in a higher ratio of rightward swims and vice versa for leftward motion. What would happen to this curve if we superimpose lateral luminance cues? Luminance is generally attractive[26,27]. Therefore, if zebrafish use an additive strategy, then we would expect that lateral luminance cues in which the right side is brighter than the left side would shift this curve upward. For lateral luminance cues of flipped polarity, we expect to see the curve shift downward (Fig. 1e). If instead fish follow a winner-takes-all strategy, responses should match the only-motion curve for strong coherence levels but switch to following lateral luminance cues at low coherences (Fig. 1f). We fitted a sigmoidal curve with additive or winner-takes-all inputs[8] (Methods) to make quantitative predictions for both these scenarios. Our behavioral experiments and model-fitting results (Fig. 1g) showed that, on average, zebrafish follow the additive algorithm.

It may be possible that the observed effects originate from combining animals that employ distinct strategies, where some fish favor motion while others rely more on luminance. To test this idea, we fitted our multifeature model to individual fish. We fitted the additive and winner-takes-all version of our model, as well as the motion-only and luminance-only versions, to account for the possibility of some individual fish relying on motion and others on luminance. While individuals exhibited variability in their response curves, the additive model described behavior better than the winner-takes-all model and single stimulus model-variations (MOT: only-motion, LUMI: only-luminance) in 76 out of the 79 tested larvae (Fig. 1h,i).

To further explore which sensory tuning parameters, such as decision-threshold and sensitivity, changes upon combining the two stimuli, we performed an additional analysis using the threshold and width parametrization of the psychometric curve[36,37] (Methods). This analysis revealed that superimposing lateral luminance cues to the motion stimulus only shifts the decision threshold while leaving coding sensitivity intact (Supplementary Fig. 1a−c), further corroborating that both stimulus features are independently processed.

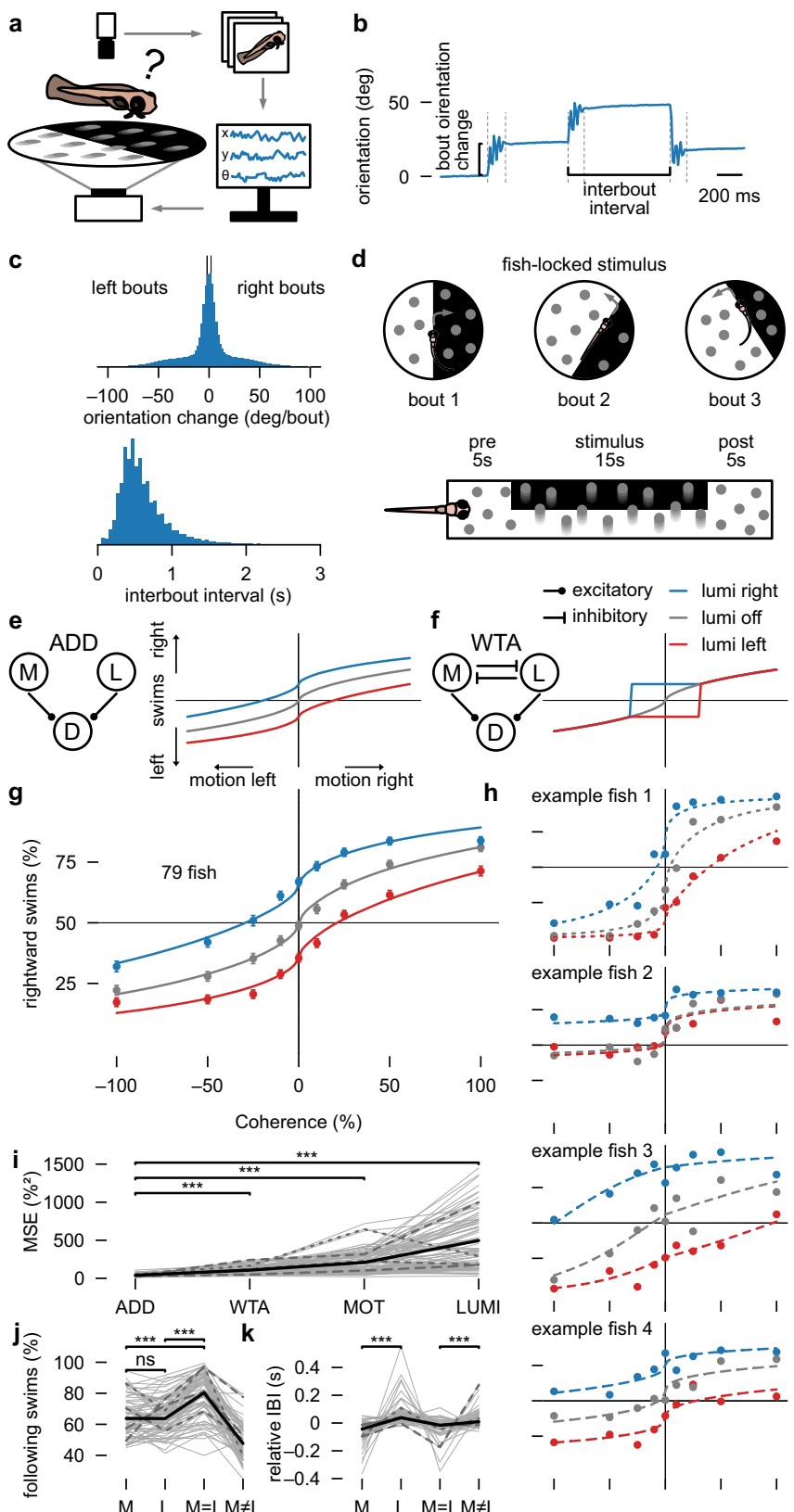

To further quantify behavior across individuals, we placed stimuli into four categories (Fig. 1j, only-motion, M; only-luminance, L; congruent motion and luminance, M = L; and conflicting motion and luminance, M ≠ L). We picked the 25 % coherence level for the motion stimulus, since at this level, motion and lateral luminance had similar directional effects on behavior. Swims in the direction of the stimulus occurred at around 65 % in the single stimulus cases (same for M or L, as expected from our stimulus choice). Performance increased to 80 % in the congruent case (M = L) and dropped to chance levels when conflicting (M ≠ L), as predicted by our additive model. Interbout intervals further provided us with a proxy of reaction time and allowed us to quantify response delays for the different stimulus configurations

**Fig. 1 | Larval zebrafish use an additive algorithm of multifeature cues to guide sensorimotor decision-making. a** Freely swimming behavior setup. Individual larval zebrafish swim in a dish while being tracked. Position and orientation are computed in real-time and used to update the visual stimulus projected from below. **b** Example raw data. **c** Normalized distributions of bout orientation change and interbout interval across all fish. **d** Stimulus structure: the visual stimulus is locked to the position and orientation of the fish, and consists of a pre, test, and post stimulus phase. **e**–**h** Behavioral response curves indicating swim direction for different motion coherence and luminance combinations. Axes as in (**g**), line colors as in the legend in (**f**). **e**, **f** Prediction by additive (ADD) and winner-takes-all (WTA) models. **g** Experimental data. Markers and error bars indicate mean ± SEM across 79 fish. **h** Randomly sampled fish. **i** Model comparison for the additive (ADD), winner-takes-all (WTA), only-motion (MOT) and only-luminance (LUMI) models (Methods). **j**, **k** Percentage following swims and relative interbout interval (Methods) across stimulus types. M: only-motion, L: only-luminance stimuli, M = L: motion in direction of bright lateral luminance side, M ≠ L: motion in direction of dark lateral luminance side. $N = 79$ fish. Dashed lines in in (**i**–**k**) indicate the four example fish from (**h**). Solid black lines indicate mean across fish. Spread of the data is visualized through the individual fish data in gray solid lines. Two-sided *t*-test *p*-values (Bonferroni correction) are indicated with ***$p < 0.001$, ns $p > = 0.05$. The *p*-values in (**i**) are ADD vs WTA: 9.62e-20, ADD vs MOT: 1.37e-16, ADD vs LUMI: 2.22e-17. The *p*-values in (**j**) are M vs L: 0.74, M vs M = L: 7.83e-21, M vs M ≠ L: 1.89e-23. The *p*-values in (**k**) are M vs L: 1.28e-10, M = L vs M ≠ L: 7.88e-5. See also Supplementary Fig. 1.

(Fig. 1k). We found significantly shorter interbout intervals in the only-motion compared to the only-luminance case, even though performance was the same (Fig. 1j). The congruent and conflicting stimulus cases further enabled us to probe the algorithm of swim initiation. If motion and luminance cues could independently trigger bouts, congruent and conflicting stimuli would result in equal swim rate increments compared to baseline. Conversely, if motion and luminance converge into a common unit to then jointly control swimming, congruent stimuli would reach the swim initiation threshold faster than conflicting ones, resulting in shorter interbout intervals. We found significantly shorter interbout intervals for congruent than for conflicting stimuli (Fig. 1k), supporting our latter hypothesis.

Thus, our findings suggest that larval zebrafish use an additive strategy to integrate coherent motion and lateral luminance cues to guide multifeature sensorimotor decision-making. We found no evidence for a winner-takes-all algorithm in this context. Furthermore, we conclude that motion and luminance cues jointly control swim initiation, leading to delayed responses in conflicting conditions as measured by interbout interval. As these conclusions are based on an analysis of steady-state behavior, we next sought to examine responses on a finer temporal scale.

## A three-pathway model captures behavioral dynamics to multifeature stimulus configurations

The difference in interbout interval for motion- and luminance-driven behavior (Fig. 1k), despite similar overall accuracy (Fig. 1j), suggests distinct processing across the two visual stimulus features. To further explore these underlying mechanisms and expand our model across the temporal domain, we next investigated how coherent motion and lateral luminance cues are processed over time. To this end, we applied a moving average window on binarized swimming direction across trials, which allowed us to compute how the percentage of leftward swims evolves along stimulus presentation.

When we presented leftward only-motion stimuli, fish mostly turned to the left (Fig. 2a). Upon stimulus onset, the percentage of turns towards the left slowly increased. Upon stimulus offset, the percentage of turns towards the left slowly decayed back to chance levels. The slow in- and decrease are consistent with previous findings of temporal integration of coherent motion cues[16,17]. Only-luminance stimuli elicited a more complex temporal response as compared to the only-motion stimuli (Fig. 2b). The fish showed a slight long-term preference for the brighter hemisphere (see the second half of the shaded area in Fig. 2b). This is consistent with previous results[26,38,39]. We also observed transient peaks at both stimulus onset and offset (Fig. 2b and ref. 27). These transient responses followed a relatively simple pattern: In cases where the arena was white before and after the stimulus, the fish tended to swim towards the white half of the lateral luminance cue. Conversely, when the arena was black before and after the stimulus, animals were more likely to swim towards the black half of the lateral luminance cue. When we presented motion and lateral luminance cues superimposed (Fig. 2c,d), we found that the behavior largely reflected

the linear sum of responses to either cue alone. Based on these results, we hypothesize that larval zebrafish employ at least three parallel computational pathways that process visual cues during sensorimotor decision-making: (1) temporally integrated motion (named motion pathway), (2) lateral luminance levels (named luminance level pathway), as well as (3) lateral luminance change (named luminance change pathway). Since the fish mostly swim in the direction of coherent motion, and display a steady-state tendency to turn towards the brighter side, we consider visual motion cues and steady-state bright lateral luminance cues to be attractive. During the transient phase of luminance change, fish swim away from changes in luminance level. Therefore, sudden changes in lateral luminance seem to be repulsive to larvae. Finally, we propose that the output of these computational pathways converges without explicit mutual interactions to drive directional swimming behavior.

To formalize this hypothesis, we developed an algorithmic model (Fig. 2e) incorporating a weighted sum of these three processing pathways. We implemented the luminance change pathway as a sign-invariant derivative, modeled as two rectifying units that compare the current luminance level with its temporally integrated form (Fig. 2e, scheme on the right). The model includes seven free parameters: a weight and a time constant per processing pathway, and one additional time constant for a multifeature integration hub where pathways converge.

We fitted our model to experimental data and validated results through a leave-out-and-testing approach. We designed a set of 40 stimuli in which motion and luminance varied across several spatial and temporal configurations (Fig. 2f, Supplementary Fig. 2a, and Methods). Stimuli classes are only-motion (M), only-luminance (L), congruent (M = L), or conflicting (M ≠ L). We fitted our model to the behavioral dynamics in response to a subset of 20 stimuli across all four stimulus classes (multifeature training, Fig. 2f). We then used the remaining 20 stimuli (multifeature testing) to validate the quality of the fit by computing the mean-squared error between model prediction and experimental data. Our analysis revealed that the experimental data closely matched the model prediction (Fig. 2g,h).

We next probed whether nonlinear interactions across the three different processing pathways in our model may influence behavior. If no nonlinear interaction exists, then a model trained separately on only-motion and only-luminance trials should be able to faithfully predict responses to combined stimuli. In contrast, if major crosstalk occurs, then this unifeature training should result in significantly worse predictions for the combined stimuli. We trained our model on 20 stimuli selected from the only-motion (M) and only-luminance (L) classes (unifeature-training, Fig. 2f). Notably, this unifeature-trained model performed just as well as the multifeature-trained model (Fig. 2g,h and Supplementary Fig. 2b), further corroborating the independent processing of visual information across the three computational pathways.

To check for the relative importance of each pathway, we performed in silico silencing simulations. To this end, we systematically

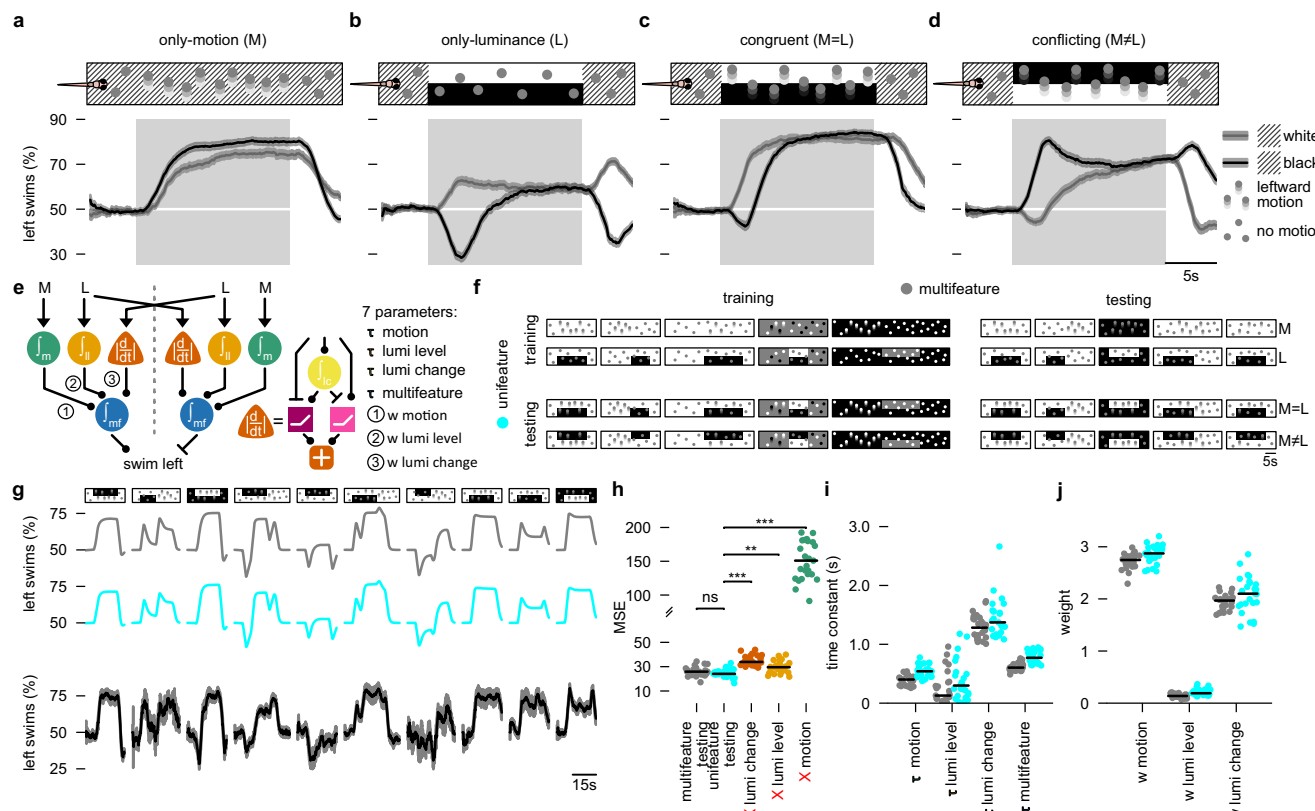

**Fig. 2 | An additive three-pathway model integrating motion, luminance, and luminance change captures multifeature visual processing. a–d** Percentage leftward swims over time for only-motion (M), only-luminance (L), congruent (M = L), and conflicting (M ≠ L) stimuli. Solid gray and black curves with shaded areas represent the mean responses ± SEM across trials with white and black backgrounds, respectively. Diagonal lines in the stimulus icons indicate when the background was either black or white. N = 64 fish. **e** Proposed parallel additive model. Green circle node: motion integrator; golden circle node: luminance level integrator; orange triangle node: luminance change detector; blue circle node: multifeature integrator; Yellow circle: another luminance integrator; purple square node: luminance decrease detector; pink square node: luminance increase detector. **f** Stimulus overview used for model testing. 40 different stimuli (Supplementary Fig. 2a) covering the four combinations of superimposed motion and lateral luminance cues. The multifeature model fitting approach splits the stimuli into training (left columns) and testing (right columns). The unifeature model fitting

approach splits the stimuli into training (top rows) and testing (bottom rows). **g** Example model predictions to 10 randomly selected stimuli not used in either multi- or unifeature training. Model predictions in gray and cyan, experimental data in black: mean ± SEM across N = 31, 13, 11, 16, 24, 16, 13, 31, 24, 11 larvae, for each stimulus, respectively. **h** Mean-squared-errors (MSE) for the full model (multifeature testing and unifeature testing) and for models lacking either the luminance change pathway, the luminance level pathway, or the motion pathway. ***p < 0.001, **p < 0.01, ns p > =0.05 (two-sided t-test; p-values are adjusted for multiple testing using the Bonferroni correction. The p-values are multifeature vs unifeature: 0.120, unifeature vs X lumi change: 8.38e-12, unifeature vs X lumi level: 1.77e-3, unifeature vs X motion: 2.47e-30. **i** Parameter distributions of the extracted time constants for each integrator node. **j** Parameter distributions of the estimated weights. Black line in **h–j** indicates the median across N = 25 fit iterations. Spread of the data is visualized by the individual fitting iterations (dots). Colors in (**g–j**) match the multi- and unifeature labels in (**f**). See also Supplementary Fig. 2.

switched off individual computational pathways in our model (Fig. 2h). Removing either the lateral luminance change or the lateral luminance level pathway had small but significant effects, reducing the quality of the fit. Removing the motion integration pathway strongly reduced model performance. We thus conclude that all three parts of our model are required to explain the observed behavioral dynamics.

With our fitting approach, we obtained reliable estimates of the seven parameters (Fig. 2i,j): The time constants for the three parallel processing pathways and the multifeature integration hub had a similar order of magnitude, with the luminance change pathway time constant being slightly higher (Fig. 2i). In agreement with our model unit silencing tests (Fig. 2h), motion carried the strongest weight (Fig. 2j). The weight for the lateral luminance level pathway played a positive, but relatively minor role. The weight for the lateral luminance change pathway was close to the weight for the motion pathway. The high weight of this pathway but the observed modest effect when silenced can be explained because the lateral luminance change pathway is only active during the short time window at stimulus transitions (Supplementary Fig. 2c).

In summary, our analyses of the behavioral dynamics across the entire period of stimulus presentation support an algorithm involving the parallel integration of three visual streams: temporally integrated directional motion cues, lateral luminance levels, and lateral luminance change. While motion and bright lateral luminance cues are attractive, larvae are repelled from the side at which luminance changes occur. A simple generative model that adds and convergently integrates these processes could successfully predict behavioral response curves for a large variety of superimposed motion and lateral luminance stimuli. Having established this computational framework, we next investigated brain-wide neural activity to identify circuits that correspond to the nodes of our model.

**Model-predicted activity is represented across the brain**
Since our model accurately predicted behavioral dynamics to a wide range of stimulus configurations, we next sought to determine whether and where its computational components may be represented in the larval zebrafish brain. To achieve this, we performed two-photon calcium imaging in *elavl3:H2B-GCaMP8s* larvae (Fig. 3a and

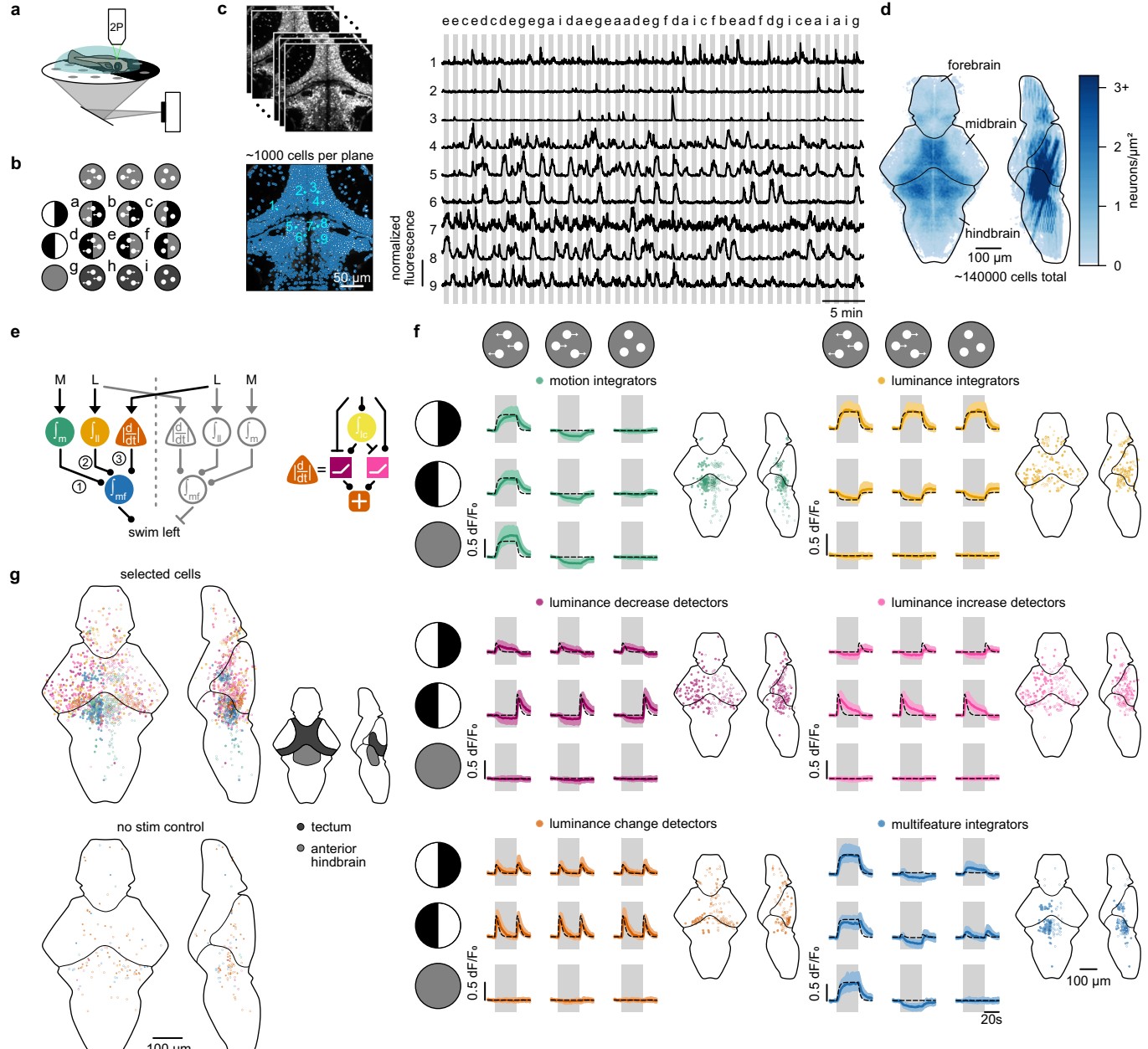

**Fig. 3 | Neural representations predicted by the parallel additive model are present in the brain. a** Two-photon calcium imaging of embedded *elavl3:H2B-GCaMP8s* larval zebrafish with visual stimuli presented from below. **b** Stimuli a–i consisted of nine combinations of motion (left, right, off) and luminance (left, right, off). Motion-off stimuli show flickering dots of 0 % coherence. Note that stimuli are illustrated in grayscale but were displayed in red (Methods). **c** Example imaging planes (top) with automated cell segmentation (bottom). Scale bar is 50 μm. Raw fluorescent traces of 9 example cells (location indicated in cyan). Y-axis scale bar indicates 1 normalized fluorescence. Letters above the traces indicate stimuli as in (**b**). **d** Overview of all imaged and segmented neurons as heatmap; the shade of blue is proportional to the number of neurons. **e** Model overview, as in Fig. 2e, highlighting only the left hemisphere through solid colored circles. Colors match the legend in (**f**). **f** Neural activity of neurons found by logical statements matching each model node highlighted in (**e**). Solid colored lines and colored shaded areas

indicate median and quartile range. Black dotted lines are the model predictions. Gray shaded backgrounds indicate when the stimulus was on. Pre- and post-stimulus consisted of 0% coherence flickering dots. Brain maps indicate the location of the neurons. Solid circles are neurons matching the left model hemisphere, open circles are neurons matching the right hemisphere. Functional traces are combined data from the left hemisphere and flipped data from the right hemisphere. **g** Overview of the brain maps containing all neurons found in (**f**). In the control (bottom), we applied our logical statements to randomly selected trials of stimulus i (0 % coherence and luminance off) See Supplementary Fig. 4c for functional traces of the control. The brain region cartoon on the right shows the tectal periventricular layer, in which most luminance processing neurons are located, and the superior dorsal medulla oblongata stripe 1–3 of the anterior hindbrain, in which most of the motion and multifeature integrators are located. $N = 15$ fish in (**d**–**g**). See also Supplementary Figs. 3–5.

Supplementary Movie 2). This transgenic line expresses GCaMP8s in the nucleus of almost all neurons of the brain. We presented combinations of coherent motion with and without superimposed lateral luminance cues (Fig. 3b). By scanning different brain regions across

individuals, we achieved near-complete coverage of the brain (Fig. 3c,d).

Our behavior-constrained model predicts the activity of seven functional nodes (Fig. 3e): One node temporally integrates global

visual motion drift. Another node integrates luminance in the luminance level pathway. We hypothesize that the node for the luminance change pathway is composed of four sub-nodes that together implement the computation of a sign-invariant derivative: An integrator of luminance, followed by separate rectifying decrease and increase detectors that converge into a luminance change detector. The decrease and increase detectors obtain their properties by comparing the integrated luminance with the momentary value. The final node, a multifeature integrator, convergently combines the three pathways. We do not expect that calcium imaging can distinguish between the integrators in the luminance level and luminance change pathways, whose time constants predicted by our model are 0.4 vs. 1.5 s, respectively (Fig. 2i). GCaMP is likely too slow to catch such subtle differences, in particular when localized to the nucleus[40]. Hence, for our brain-wide analysis, we treated the two luminance integrators as one unit, putting the total of functional nodes to six.

We detected neurons that match the model nodes using logical statements based on the temporal activation patterns predicted by our model (Methods). This analysis revealed a map of neural representations matching the model nodes. We observed a hemispheric symmetry in which leftward driving stimuli are mostly represented in the left hemisphere and vice versa for rightward driving stimuli (filled vs. open circles in Fig. 3f,g). Specifically, we found luminance cues primarily represented in the optic tectum (Fig. 3f,g and Supplementary Fig. 3a), with additionally a high density of luminance integrators in the habenula (Supplementary Fig. 3a). To probe if the luminance integrators can truly act as integrators, we also tested different contrast levels. This experiment indicated that neural activity rises more slowly in cases of lower contrasts compared to higher contrasts, adding confidence to their capability to temporally integrate visual cues (Supplementary Fig. 3b,c). Motion and multifeature integration occurred predominantly in the anterior hindbrain (Fig. 3f,g and Supplementary Fig. 3a). These representations are consistent with the projection patterns of visual inputs from the eye to the contralateral optic tectum[41], from the eye through the eminentia thalami to the left dorsal habenula[34,42] and from the pretectum to the ipsilateral anterior hindbrain[25,35]. Using a complementary classical regressor-based correlation analysis (Methods), we qualitatively confirmed these spatial arrangements (Supplementary Fig. 4a,b). With both methods revealing similar results, we wanted to know if our logical statement-based approach may have benefits compared to a regressor-based analysis. To test this, we generated synthetic neural data of two model units and added different levels of noise to the traces (Methods). We then checked to what degree either strategy could reveal unit identity. Our logical statement-based analysis outperformed regressor-based correlation analysis in low signal-to-noise-ratio regimes, indicating a clear benefit when signals are transient or noisy (Supplementary Fig. 4e–h). We also sought to control for potential false-positive assignments, where our methods may label neurons by chance without reliable stimulus-induced activity. To this end, we applied both analyses on periods during which neither motion nor lateral luminance cues were present (Fig. 3b, flickering dots on a dark background = stimulus i). As expected, for both analysis methods, we labeled only a few neurons across the brain, without any spatial arrangement (Fig. 3g and Supplementary Fig. 4c).

Our behavioral data support an additive integration strategy over a winner-takes-all algorithm (Fig. 1e–g). This led us to ask whether winner-takes-all dynamics may be completely absent in the brain. In the case of winner-takes-all, contralateral motion and luminance integrators would inhibit each other to solve conflicts. The integrator with the strongest input would win and silence the weaker integrator. In the case of similar inputs, one of the integrators would randomly take over (Supplementary Fig. 4d). Using our logical statement-based classification approach (Methods), we found such cells to be present across the brain. However, compared to neurons matching our additive

model, there were fewer cells with such dynamics with less prominent spatial arrangement (compare Fig. 3f and Supplementary Fig. 4d). These winner-takes-all cells may thus play a less important role in the sensorimotor computations under the conditions we tested in our assays. They may be recruited in different contexts, such as more complex tasks, time-constrained decisions, or at later developmental stages.

Our functional imaging results provide neural evidence that our proposed behavior-constrained model (Fig. 2e) could indeed be mechanistically implemented on the level of the nervous system. The optic tectum emerged as a hub for detecting lateral luminance changes, while the anterior hindbrain appeared as a key site for motion integration and multifeature sensory processing. These findings suggest a structured, spatially organized circuit in which motion and luminance information are processed in parallel and then linearly added to guide behavior.

## Better separability of congruent than conflicting stimuli in low-dimensional space

Does an unsupervised dimensionality reduction approach yield similar conclusions to our model-based predictions? To address this question, we performed a neural manifold analysis using principal component analysis (PCA) on single imaging trials (Supplementary Fig. 5a). We found that 30 principal components were necessary to explain 53% of the total variance (Supplementary Fig. 5b). The number of necessary components is likely so high because our dataset aggregates neural recordings across multiple fish, with each fish imaged in different locations across the brain and being presented with a different random trial order. Nevertheless, given the constrained stimulus space of only nine combinations of motion and luminance, we hypothesized that key stimulus features would readily emerge in the leading principal components. Indeed, when projecting brain-wide activity into three principal dimensions, we observe a separation between two different axes corresponding to motion and lateral luminance processing (Supplementary Fig. 5c). We computed the Euclidean distance in this lower dimensional space between opposing stimulus directions. This analysis revealed that left- and rightward stimulus conditions diverged more rapidly when motion and luminance are congruent than when they are in conflict (Supplementary Fig. 5d), suggesting better separability of directions when cues are aligned. This increased separability might play a role in the faster decision times we observed in our behavioral experiments for congruent stimuli (Fig. 1k). When we split the data into the major brain regions (fore-, mid-, and hindbrain), we found most rapid divergence in the presence of luminance in the midbrain, as well as increased separability of congruent as compared to conflicting stimuli in the hindbrain (Supplementary Fig. 5c,d). Together, these findings revealed coarse brain-wide spatiotemporal encoding patterns for motion and lateral luminance stimuli and tie into the observed faster decision times in our behavior experiments.

## Excitation and inhibition are largely balanced across motion and lateral luminance processing neurons

Through our behavior experiments, algorithmic modeling, and imaging experiments, we proposed a parallel additive model of sensorimotor decision-making whose computational units are spatially arranged in the larval zebrafish brain. These experiments do, however, not allow constraining the sign of model connections. To address this problem, we performed hybridization chain reaction fluorescent in situ hybridization (HCR-FISH) (Fig. 4a–c). After functional imaging, we fixed and stained larvae for *gad1a*, *gad1b*, *gad2*, *vglut2a* and *vglut2b*. Probes against *gad* variants label GABAergic, presumably inhibitory, cells. Probes against *vglut* variants label glutamatergic, presumably excitatory, neurons. After staining, we then imaged larvae again, computationally aligned the two imaging stacks at cellular resolution and manually classified the neurotransmitter identity (Supplementary

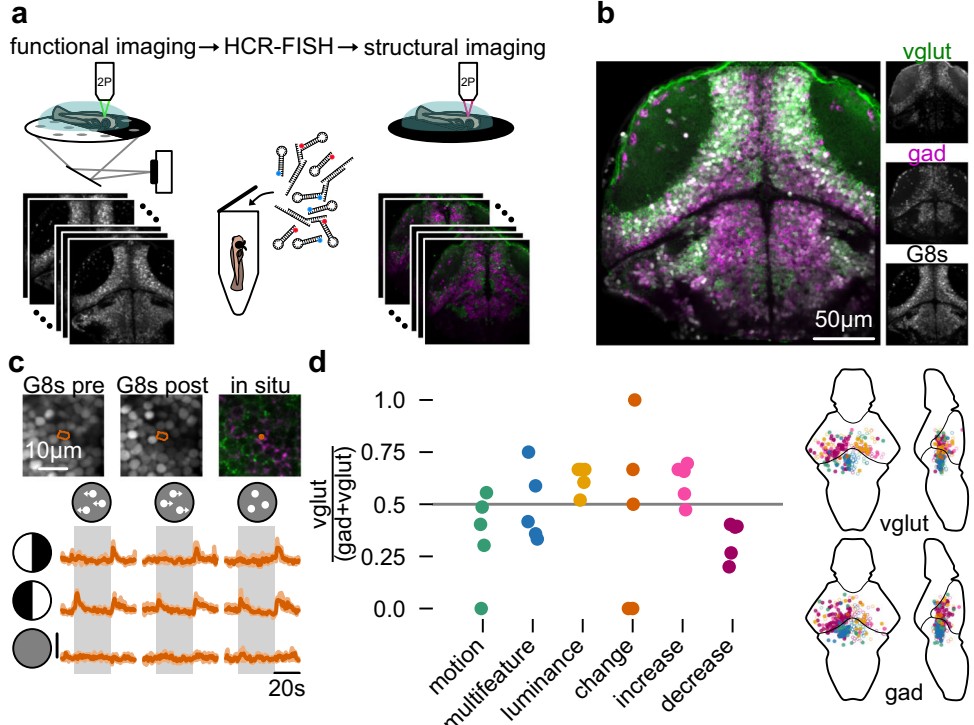

**Fig. 4 | HCR-FISH reveals a balance of excitation and inhibition across computational units. a** HCR-FISH pipeline. After functional imaging, we fixed and stained the fish for *gad* and *vglut*, and then imaged the fish again. **b** Overview of *gad* (using combined probes for *gad1a*, *gad1b*, and *gad2*) and *vglut* (using combined probes for *vglut2a* and *vglut2b*) labeling on top of H2B-GCaMP8s (G8s) background in an example fish. **c** Example luminance change detection neuron from one example fish with its functional traces. Images on top show the H2B-GCaMP8s channel before (G8s pre) and after (G8s post) HCR-FISH treatment, as well as the gad/vglut overlay (in situ). Colors of the in situ image as in (**b**). Fluorescence scale bar indicates 0.5 dF/F$_0$. Solid curve and shaded areas indicate median and quartile range across $N = 8$ trials for stimuli a, b, e, f, g, and i. $N = 9$ trials for stimuli c and h. **d** Left: Ratio of excitatory neurons (vglut/(gad + vglut)) across the six functionally identified cell types in $N = 5$ fish. Luminance integrators had a tendency to be slightly more excitatory (two-sided *t*-test compared to 0.5 baseline; $p = 0.15$) and luminance decrease detectors a tendency to be slightly more inhibitory (two-sided *t*-test compared to 0.5 baseline; $p = 0.17$). Right: Location of the excitatory (top) and inhibitory (bottom) neurons. See also Supplementary Figs. 6 and 7.

Fig. 6a–c). Across all cell types, we found mostly balanced populations of excitatory and inhibitory cells (Fig. 4d), a common circuit motif for temporal integration[43]. We observed a tendency for the population of luminance integrators and luminance increase detectors to be slightly more excitatory. Luminance decrease detectors seemed to be slightly more inhibitory. We grouped cells based on neurotransmitter type, revealing no obvious differences in their functional activity (Supplementary Fig. 7a). This result suggests that both excitatory and inhibitory neurons are part of the same computation, further corroborating the idea of balanced neurotransmitter ratios. To further refine our behavior-based model, we still require morphological characterization of the projection patterns across the brain.

### Neuron morphology of computational units refines model structure

Our behavioral experiments allowed us to propose a relatively simple model structure using computational nodes (Fig. 2e). Through our imaging experiments, we found neural elements that may implement the underlying computations (Fig. 3f,g). Our neurotransmitter analyses revealed a largely balanced circuit arrangement (Fig. 4d). To further constrain and refine our model, we next sought to investigate neuronal morphology. We performed two types of analyses, broadly within the identified brain areas as well as more targeted to the functionally labeled cell types.

First, we constrained potential anatomical architectures using morphological data downloaded from the mapzebrain atlas, a publicly available reference platform for larval zebrafish multimodal neural data[44]. Binarized kernel density estimation (KDE) maps of our functionally identified neurons (Fig. 3f) revealed overlapping regions for motion, multifeature integrators, and the four luminance-processing types. We merged these maps into one tectal and one anterior hindbrain map (Supplementary Fig. 8a). We then selected neurons with somata within these regions (Methods) and assigned them to four possible tectum–anterior hindbrain pathways (Supplementary Fig. 8b): (1) Ipsilateral tectal projections reaching ventrally projecting hindbrain neurons. (2) Ipsilateral anterior hindbrain projections connecting with ventrally projecting tectal neurons. (3) Contralateral hindbrain projections receiving input from ventrally projecting tectal neurons. (4) Contralateral tectal projections reaching ventrally projecting hindbrain neurons. We found all four anatomical arrangements to be present in the larval zebrafish brain. The diversity of cell type projections between the tectum to the anterior hindbrain highlights the need for more targeted approaches that combine structure and function analysis within the same preparation. Our atlas-based results provided, however, important evidence about the order of magnitude and location of neurons that we needed to generate via such relatively complex experiments.

To obtain functional and structural information simultaneously from the same neurons in the same animal, we performed functionally guided pa-GFP photoactivations[35,45] (Fig. 5a, Supplementary Fig. 8c,d, Supplementary Movie 3, and Methods). We first selected individual neurons based on their functional activity. With a targeted laser pulse, we then photoconverted cytoplasmic pa-GFP, which then distributed across the whole cell, allowing us to trace the morphology of our

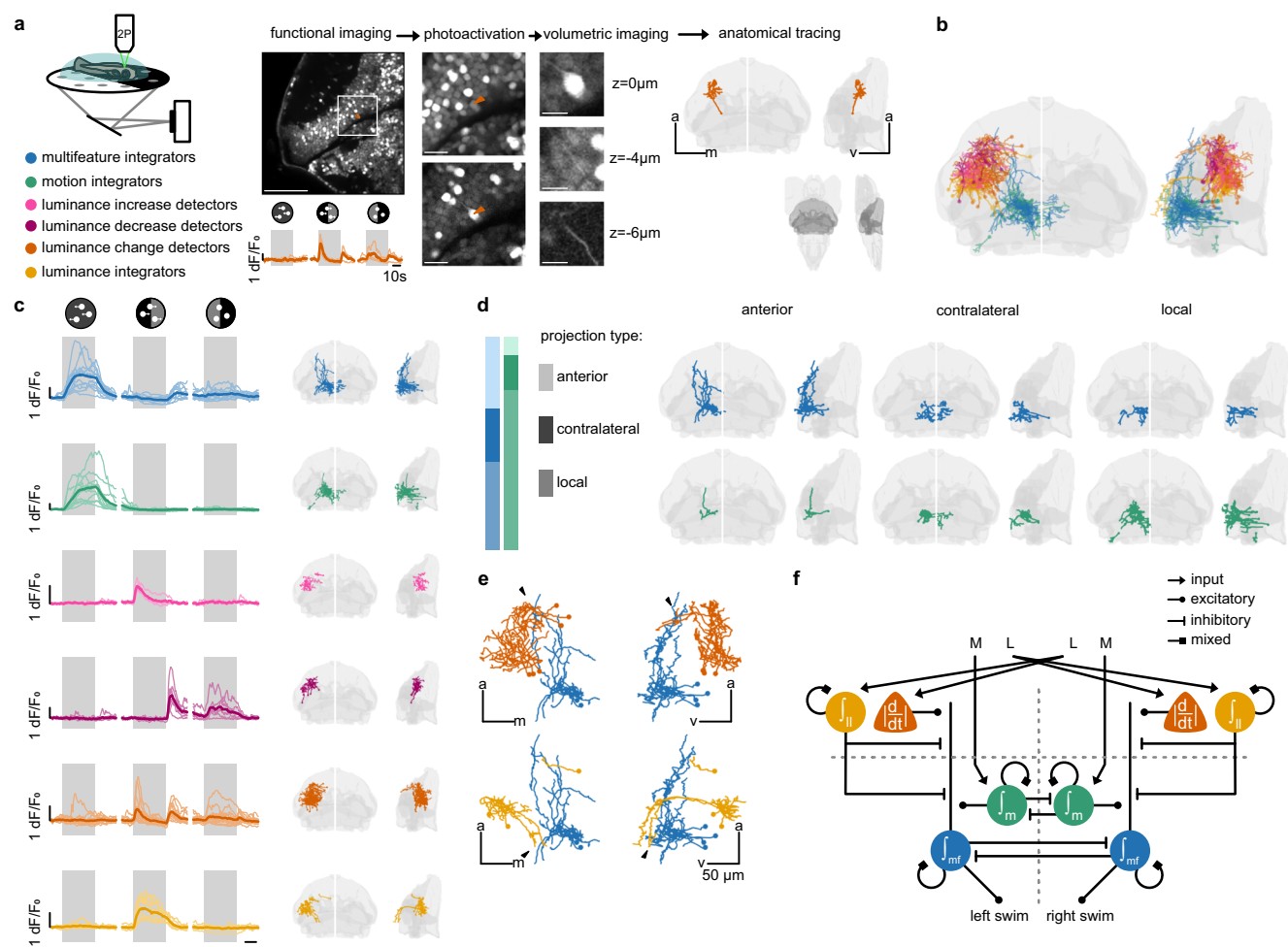

**Fig. 5 | Projection patterns of functionally identified cells refine our model.**
**a** Functionally targeted pa-GFP photoactivation pipeline with an example luminance change detector neuron. This example is representative for all 63 photoactivated neurons shown in **c**. Left 2 P image: Functional imaging and manual cell type identification based on a minimal stimulus set containing three stimuli. Scale bar is 50 μm. Middle 2 P image: Cell before photoactivation (top) and after (bottom). Images match the white-square in the overview image (left). Scale bar is 10 μm. Right 2 P image: Volumetric imaging after photoactivation. Scale bar is 5 μm. Right brain map: Display of reconstructed neuron morphology. Reference brain regions in light gray: optic tectum, anterior hindbrain sup. dMO stripe 1, and cerebellum; as highlighted in dark gray in the full brain overview below. **b** All 63 traced neurons of all functional types. Colors related to legend in (**a**). **c** Functional traces and morphologies across cell types. Gray shaded background indicates when the stimulus was on. From top to bottom: 12 multifeature integrator neurons

(blue), 12 motion integrator neurons (green), 4 luminance increase detector neurons (pink), 7 luminance decrease detector neurons (purple), 18 luminance change detector neurons (orange), and 10 luminance integrator neurons (yellow). Reference brain regions as in (**a**). **d** Proportion and morphology of local, anterior, and contralateral projections of multifeature (blue) and motion (green) integrator neurons. The multifeature integrator morphologies (top row) contain 4 anterior, 3 contralateral, and 5 local projections. The motion integrator morphologies (bottom row) contain 1 anterior, 2 contralateral, and 9 local projections. **e** Top: Close-up of potential connections between luminance change detectors (6 neurons) and anterior projecting multifeature integrators (4 neurons). Bottom: Close-up of potential connections between luminance integrators (3 neurons) and anterior projecting multifeature integrators (4 neurons, same as on top). **f** Refined circuit model. See also Supplementary Fig. 8.

selected neuron. We mapped reconstructed cells into a common reference frame and compared neuronal morphologies across animals. We generated a library of 63 cells for which we know their activity profile as well as their morphological structures (Fig. 5b,c and Supplementary Movie 4). With this library, we could also speculate about potential connectivity across cell types based on proximity.

We found the following projection patterns of the six functional types (Fig. 5c,d and Supplementary Fig. 8e): The multifeature neurons project either into the interpeduncular nucleus (IPN), anteriorly into the hypothalamic region of the forebrain, or contralaterally to the other side of the anterior hindbrain, with one exception remaining local. Most motion integrators in the anterior hindbrain project locally and remain ipsilateral, largely intermingled with the neurites of the multifeature neurons. The identified luminance increase detectors all remain local in the tectum and mostly project to the stratum fibrosum

et griseum superficiale (SFGS). The luminance decrease detectors also remained mostly local within the tectum, with a majority of cells projecting to the stratum album centrale (SAC) and stratum griseum centrale (SGC). Among the luminance change detectors, we find several projections towards deeper tectal neuropil layers, as well as projections into the forebrain, specifically the pretectum, dorsal thalamus, and intermediate hypothalamus. These ventral projections reach proximity to the anterior side of the neurites of multifeature neurons (Fig. 5e, top). For the luminance integrators, we largely find the same projection patterns. However, notably, projections from this cell type arrive at the posterior side of the neurites of multifeature neurons (Fig. 5e, bottom).

Our combined functional and structural analyses allow us to refine our behaviorally constrained model (compare Fig. 3e and Fig. 5f). We justify the arrangement of the first processing layer with the following

findings: In our imaging experiments, we found that lateral luminance cues are represented mostly in the optic tectum on the contralateral side (Fig. 3f). We thus propose a connectivity arrangement in which lateral luminance signals from the right eye get processed in the left hemisphere and vice versa. In our photoactivation experiments, we found luminance increase and decrease detectors to have largely local projections within the tectum (Fig. 5c and Supplementary Fig. 8e). We therefore suggest that the computation of luminance change happens locally within the optic tectum. Motion integrators remained mostly local in the anterior hindbrain in our photoactivation experiments (Fig. 5c and Supplementary Fig. 8e). We modeled these cells accordingly.

We argue for the second processing layer of our refined model as follows: Based on the observed anatomical spatial arrangement of luminance integrator cells (Fig. 5e, bottom), we modeled them to have ipsilateral connections to multifeature neurons. Luminance integrator cells receive only input from the contralateral side in the first processing layer of our model. This means if they would be excitatory, multifeature neurons should display sustained activity for contralateral luminance cues. As this, however, is not the case (Fig. 3f, stimulus f), we modeled luminance integrator cells to likely inhibit multifeature neurons. Luminance change detectors project ipsilaterally onto multifeature neurons (Fig. 5e, top). As luminance changes on the contralateral side induce transient responses in multifeature neurons (Fig. 3f, stimulus f), we implemented these cells to excite multifeature neurons within the same hemisphere. Motion integrators and multifeature integrators both display sustained activity when activated by coherent motion (Fig. 3f, stimulus g). As we found mostly local projections for motion integrators within the anterior hindbrain (Fig. 5d), we modeled these cells to provide excitation to multifeature neurons. Each of the three processing pathways project onto spatially distinct compartments of multifeature neurons in our model. We speculate that such differential targeting might suggest potential mechanisms of dendritic computations within these cells.

We further argue for projections across hemispheres: In our photoactivation experiments, we identified contralaterally projecting motion integrators and multifeature neurons (Fig. 5d). Previous work in our lab showed that the vast majority of contralaterally projecting motion-sensitive neurons in the anterior hindbrain are gad-positive[45]. Therefore, we model these contralateral connections as likely inhibitory. These connections may explain the slight decrease in activity for motion stimuli in the non-preferred direction in both the motion and multifeature integrator neurons (Fig. 3f, stimulus b, e, and h).

Finally, we propose recurrent connectivity within each of the integrators: We argue for this arrangement based on our HCR-FISH results that showed largely mixed excitatory and inhibitory populations (Fig. 4d). This balance is likely required for the implementation of the temporal integration properties well beyond the single-neuron timescale within each computational unit.

In summary, we have used a variety of complementary methods, involving behavioral experiments, functional imaging, HCR-FISH, and single-cell pa-GFP photoactivation to test and constrain a model of multifeature sensorimotor decision-making. The proposed model agrees with an additive strategy and has its computational nodes located in the tectum and anterior hindbrain.

## Discussion

Our study demonstrates that larval zebrafish integrate multiple visual features—motion, luminance level, and changes in luminance—through an additive strategy to guide sensorimotor decisions. We identify a distributed neural representation of these computations and propose a plausible circuit-level organization in which distinct stimulus features are processed in parallel and subsequently converge to drive behavior.

The optomotor response has been extensively characterized as a spatial and temporal integration process[16,17,23,46]. Visual motion is encoded retinotopically in direction-selective layers of the tectum[47] and in the pretectum[25,35,48], and then further processed[25] and temporally integrated in the anterior hindbrain to regulate motor behavior[16,17]. We also observed temporally integrated motion responses in the anterior hindbrain, alongside motion-responsive neurons in both tectal and pretectal regions. Our multifeature stimulation approach enables us to further segregate functional types into neuronal populations that respond selectively to motion or luminance, or to both of these cues.

To assess lateral luminance-guided behavior, we adapted a paradigm in which lateral luminance cues are locked to the position and body orientation of the fish in real-time[27,31,39], allowing us to quantify phototaxis in a precisely controlled stimulation arrangement. As we could show previously[27], phototactic behavior consists of two components: a sustained attraction to bright stimuli and a short-latency repulsion from abrupt lateral luminance changes. The representation of relevant visual signals, like changes in brightness or luminance level, has been described in retinal and early sensory processing regions[49,50]. Work on the downstream phototaxis circuitry in zebrafish implicates the preoptic area[51], habenula[34,42], and thalamus[52]. In our imaging experiments, we did not find a contribution of the preoptic area, making a dominant role of this brain region in our paradigm unlikely. In the habenula of some fish, we identified luminance integrators and luminance change detectors, but no luminance increase or decrease detectors. For the thalamus, we confirmed the presence of luminance-sensitive cells. While we decided to focus our analysis on the optic tectum, where we observed all four luminance processing types, we do not rule out alternative processing pathways involving the habenula or thalamus.

Our results further suggest that motion and luminance cues are processed largely in parallel up until the anterior hindbrain. The separation of visual features in the retina and early processing regions is expected based on previous work[49,53,54] and should support flexible context-dependent analyses of visual scenes. The existence of parallel streams is corroborated by findings that indicate that color differentially affects the optomotor response and phototaxis[55] and that the ablation of phototaxis-related retinal ganglion cells has little impact on the optomotor response[52].

Luminance-sensitive neurons have been identified in the anterior hindbrain before by light-sheet microscopy[20,21]. Using two-photon imaging, we confirmed and extended these findings, ruling out potential confounds from high-intensity blue light illumination. Many of the luminance-sensitive cells that we have found in the anterior hindbrain also respond to motion. This result suggests that luminance and motion processing pathways converge onto multifeature neurons where signals are integrated additively. Our neural circuit-based observations thus agree with our findings from our behavioral analyses that also indicated an additive processing of motion and luminance cues.

Our experiments provide further evidence that the anterior hindbrain forms a key computational processing structure where diverse sensations are integrated and transformed into motor decisions[16,18–20,22]. This brain region has also been recently attributed to represent heading direction in a navigational context[24]. However, head direction neurons by themselves are not tuned to visual motion[56]. This implies that the identified anterior hindbrain multifeature and motion integrators may serve as a parallel computational unit that controls behavior, while the heading direction system synergistically keeps track of movement decisions. These ideas align with work in the central complex in *Drosophila*, where multisensory information converges to control navigational behavior[57].

In mammals, multisensory integration has been described in the dorsal cochlear nucleus of the hindbrain[58], where auditory and somatosensory information converge. Auditory and visual inputs are additively integrated in multimodal neurons in the anterior frontal cortex[8]

and the superior colliculus[59–61], a homolog to the zebrafish optic tectum. In humans, multisensory regions like the posterior superior temporal sulcus and middle temporal gyrus exhibit modality-specific substreams that interact depending on task context[62]. Besides additive processing across stimuli, computational modeling work has suggested divisive normalization as an alternative convergent strategy during multisensory integration[63]. While modular, additive, and context-flexible integration seems to be a shared analogy across mammalian systems and our results, the exact structural implementation and brain region associations are likely distinct in different species and scales of organization.

Our behaviorally constrained model is, as far as we know, the simplest configuration to capture motion- and luminance-based sensorimotor decision-making across the tested stimulation configurations. We cannot, however, exclude that more complex variants are implemented in the brain. For example, adding mutual inhibition between the two motion integrators did not improve fits (Supplementary Fig. 2d), but the identified contralateral projections (Fig. 5d) and decreased activity of motion integrators during non-preferred motion stimuli (Fig. 3f) show that the brain implements mutual inhibition motifs (Fig. 5f). Some anterior hindbrain neurons also exhibited winner-takes-all dynamics (Supplementary Fig. 4d), even though this feature was not necessary to fit our behavioral data (Supplementary Fig. 2d). These neurons may represent computational units that are selectively engaged under different task demands, developmental stages, or internal states. For example, in a time-pressured situation when needing to escape from multiple threatening stimuli, pure additive algorithms would blur the percept and be maladaptive[11]. Here, a flexible recruitment of winner-takes-all mechanisms may provide an advantage.

Our anatomical analyses suggest a spatial organization of inputs onto multifeature neurons, consistent with the idea of dendritic compartmentalization. Such mechanisms may enhance computational capacity and enable context-dependent gating logic[64,65]. More precise connectivity analyses, likely requiring correlated light and electron microscopy or related methods[45], will be required to validate this hypothesis.

Our behavior model predicted distinct excitatory and inhibitory roles across nodes. However, our function-guided neurotransmitter profiling revealed largely mixed populations. This balance of excitation and inhibition can support slow temporal integration of input signals[43]. It may also enhance developmental robustness by maintaining stable sensorimotor function during ontogenetic circuit refinement. Future experiments combining neurotransmitter identity and anatomy within the same neurons will help refine these assignments and help to further challenge these ideas.

In our neural circuit model design, we propose connectivity based on proximity after function-guided anatomical labeling. While spatial closeness does not confirm synaptic connections[66,67], it provides a relatively simple and scalable framework to develop experimentally testable network models. It would be optimal to include perspectives from function-correlated electron microscopy to constrain models[45,67], but it may also be sufficient to use readily available anatomical reconstructions from existing datasets[45,68,69].

Future work should examine whether the additive integration observed here extends to cross-modal interactions, such as visual–olfactory[70], visual–thermal[71,72], or visual–mechanosensory[73] integration, and how these computations change with experience[74] or development[27,75]. It will be important to perform manipulation experiments via, for example, all-optical circuit interrogation methods[76], to probe the predictive power of our model and its causal links to behavior. For example, one may unilaterally target one hemisphere and probe if activation of the identified cell types generates behavioral biases that are in agreement with model predictions. Finally, emerging spatial transcriptomics approaches[77] should further

help identify molecular markers to extend the biophysical realism of our models.

In summary, our results demonstrate that larval zebrafish integrate distinct visual features through a parallel, additive process mediated by the anterior hindbrain. This circuit organization mirrors general modularity principles of vertebrate brains, providing a foundation for understanding how distributed neural systems transform sensory information from complex environments into actions.

## Methods

### Animal care and transgenic zebrafish
We housed and handled adult and larval zebrafish (*Danio rerio*) according to standard procedures. We used the following lines: wild type AB strain, *Tg(elavl3:H2B-GCaMP8s)*[mpn438], *Tg(alpha-tub:c3pa-GFP)*[a7437Tg]; *Tg(elavl3:Hsa.H2B-GCaMP6s)*[jf5Tg78];. The three transgenic lines are in a mixed AB and TL genetic background. We generated the new *Tg(elavl3:H2B-GCaMP8s)*[mpn438Tg] line by standard procedures. In brief, we injected the Tol2 vector transgene construct *Tol2-elavl3:H2B-GCaMP8s*[79], obtained from Janelia Research Campus, and transposase RNA into 1–4-cell-stage embryos. We then isolated transgenic lines by screening for high expression of bright green fluorescence in the central nervous system in the next generation.

We raised larvae in small groups (~50 individuals) in Petri dishes (14.5 cm diameter) in E3 fish water + methylene blue under a 14:10 hr light-dark cycle at 28 °C. After one day, we cleaned the dishes and raised the larvae further in E3 fish water without methylene blue until 5 days-post-fertilization (dpf). We performed all experiments with larvae at 5 dpf. Sex cannot be determined at this age. Animal housing and care was approved by the Regierungspräsidium Freiburg, Germany.

### Behavior
We used WT-AB larvae for freely swimming behavior experiments, in a similar tracking setup as we developed previously[16]. In short, we transferred individual larvae into custom-designed circular acrylic dishes (12 cm in diameter, 5 mm in height, black rim, transparent base covered with diffusion paper inside the water) filled with ~50 mL of E3 fish water. We projected visual stimuli onto the dish from below with AAXA P300 Neo Pico projectors. Infrared LED strips (940 nm panel, SOLAROX®) illuminated the dishes from below, so a high-speed CMOS camera (Basler acA2040-90um-NIR or Grasshopper3-NIR, FLIR Systems) with a zoom lens (#58-240-6X, 18-108 mm FL, Edmund Optics) and an infrared long pass filter (Linghuang Zomei IR 850 nm 52 mm) could record the larva.

A custom written, Python 3.12-based software live-tracked larvae at a rate of 90 Hz. The software determines larval position based on the largest center-of-mass after background subtraction. Next, second-order image moments define larval orientation. To identify events of high activity (bouts), the software calculates a 50 ms rolling variance window over the time-evolving orientation of animals. We defined bouts to start once the rolling window variance exceeded 1 deg$^2$ for at least 20 ms. Bouts ended once the variance dropped below 0.5 deg$^2$ for at least 50 ms. Besides the raw trajectories, the software stores the bout variables: time, x position, y position and orientation at the start and end of each bout. From these variables, we determined the orientation change per bout, distance traveled per bout, and the interbout interval (time between the starts of two consecutive bouts).

### Visual stimuli
We used combinations of lateral luminance cues with random dot motion kinematograms. Stimuli were aligned to the position and body orientation of the fish in real-time.

Lateral luminance stimuli consisted of one bright and one dark semicircle (black and white, gray and white, or black and gray) positioned on each side of the fish. The two semicircles were connected by a 0.6 cm linear transition, preventing a sharp edge directly under the

fish (see also Supplementary Movie 1). For the behavior experiments, we projected white scenes at ~1500 Lux, gray scenes at ~900 Lux and black scenes at ~10 Lux, unless indicated otherwise. For the imaging experiments, to prevent bleed-through into the green photomultiplier, we used red colors. We projected red scenes at ~2 Lux, dark red scenes at ~1 Lux, and black scenes at ~0.5 Lux. These brightness values were measured via an LX1330B light meter (Hongkong Thousandshores Ltd., Central Hong Kong). Before and after each lateral luminance stimulus presentation, we showed a baseline homogenous background. For the behavior experiments, Fig. 2f and Supplementary Fig. 2a indicate whether the homogenous background was black, gray, or white. For the imaging experiments, the homogeneous background was always dark red. For the imaging experiment in which we test the temporal integration properties of the luminance integrators (Supplementary Fig. 3b), we measured responses to a left-right contrast of 0.5, 0.4, and 0.3 projector units.

On top of the lateral luminance or homogenous background, we displayed a random dot motion kinematogram. These stimuli consisted of ~1000 gray or red dots, unless indicated otherwise (each dot was 2 mm in diameter), projected from below onto a circular arena (12 cm for behavior, 8 cm for imaging). Each dot had a mean lifetime of 200 ms and stochastically disappeared, then immediately reappeared at a random location in the arena. The coherence level, specified per experiment, indicates the percentage of dots that moved coherently to the left or right at a speed of 1.8 cm/s. We showed 0% coherence as a baseline stimulus (no movement, flickering dots).

During the behavior experiments, the timing of the pre-baseline, stimulus, and post-baseline varied as is indicated in the figure legends and illustrated in Supplementary Fig. 2a. We tested various combinations of motion and luminance to ensure robustness of our results. During each behavior experiment, in the first 5 s before the start of each actual stimulus, dots were moving towards the center of the dish to guide the fish away from the walls. This phase helped to reduce thigmotactic edge-effects. During the imaging experiments all stimuli consisted of 10 s pre-baseline, 30 s stimulus and 20 s post-baseline.

We rendered all stimuli online, using custom-written software based on Python 3.12 and Panda3D 1.10.15 with OpenGL Shading Language (GLSL) vertex shaders running on AMD Radeon RX 580 GPUs.

## Behavior data analysis

To preprocess our acquired freely swimming behavior data, we combined all bouts per fish per experiment. We excluded bouts and trials affected by tracking errors[27]. The following bouts were dropped: (1) bouts with an interbout interval > 10 s. This filters out tracking mistakes where the algorithm tracks small particles on the edge of the dish. (2) Bouts where the contour area of the fish exceeded 2000 pixels. This filters out tracking mistakes where accidental air bubbles or scratches got tracked instead of the fish. (3) Bouts with an average speed > 1 cm/s. This filters out tracking mistakes where the algorithm jumps from the edge or bubble back to the fish. (4) Bouts with an orientation change > 150°. This filters out tracking mistakes where the head and tail got swapped (5) Bouts that were within 0.25 cm or less from the edge of the dish. This removal avoided edge effects in our analysis. In addition, we dropped the entire trial, if more than 5% of the bouts were labeled as tracking errors in any of the aforementioned categories. In total, these filtering steps removed 13.7 ± 11.1% of bouts and 14.3 ± 10.1% of trials per experiment. We then binarized swim bouts into left or right bouts, ignoring all forward bouts between −2 and 2 degrees.

For the steady state behavior analysis (Fig. 1), we calculated the percentage of leftward bouts (left / (left + right) * 100%) during the last 10 s of the stimulus. We use only the last 10 s of the stimulus, since the percentage of leftward bouts reaches a steady state by then. We based the relative reaction time and following swims plots on the data from 25% coherence, since at this coherence level, motion and luminance

balance each other. To get the relative reaction time, we subtracted the average reaction time across stimuli (M, L, M = L and M ≠ L) per fish. For the percentage following swims (Fig. 1j), we flipped the orientation data during rightward stimuli (considering motion during motion-luminance conflicts) and then merged the right and leftward stimuli.

For the analysis of decision curves over time (Fig. 2), we flipped the orientation data as explained above. Unless indicated otherwise, 'percentage leftward' plots are based on the combined data of actual leftward stimuli and flipped rightward stimuli. We then merged all trials per stimulus and calculated the percentage of leftward bouts in a rolling 2 s window (moving in 0.1 s steps). In the decision curve plots, each window contains bouts from up to 2 s in the past.

## Additive and winner-takes-all steady-state models and fitting strategy

To determine how zebrafish integrate motion and luminance cues, we tested whether their behavior aligns with an additive model or a winner-takes-all variant of this model. Our modeling approach was based on a recent framework describing rodent multisensory decision-making[8]:

$$p(R) = \sigma\left(m_r M_r^\gamma - m_l M_l^\gamma + l_r L_r - l_l L_l + b\right)$$

Where $p(R)$ is the probability of swimming right, and $\sigma(x) = \frac{1}{(1+e^{-x})}$ is the logistic function. Motion coherence levels are represented by $M_r$ and $M_l$ (ranging from 0 to 1), while $L_r$ and $L_l$ indicate luminance cues (either 0 or 1). In the winner-takes-all variation, the weaker stimulus is set to zero. In the single stimulus variants (MOT and LUMI), the second stimulus is set to zero. Both models include six free parameters: bias ($b$), coherence exponent ($\gamma$), motion sensitivity ($m_r$, $m_l$), and luminance sensitivity ($l_r$, $l_l$). We fitted the percentage right over coherence curves of single fish and the group average to the additive, winner-takes-all, and single stimulus model variants using scipy's curve_fit function with default parameters (initial guess: $m_r = m_l = -1$, $l_r = l_l = 0.5$, $b = 0$, and $\gamma = 0.6$). We used these initial guesses to create the model hypothesis plots in Fig. 1e,f. We then use the MSE between the model fit and data to determine which model performed best (lower is better).

## Threshold-and-width-based psychometric curve fitting

To determine which sensory tuning parameter is affected by superimposing motion and lateral luminance cues, we used the threshold and width reparametrization of the psychometric curve. This version of the psychometric curve describes the tuning of the system, rather than the decision-outcomes[36,37,80,81]:

$$p(R) = \sigma\left(\frac{-2\log\left(\frac{1}{0.8} - 1\right)(M - t)}{w}\right) \quad (1)$$

Where $p(R)$ is the probability of swimming right, and $\sigma(x) = \frac{1}{(1+e^{-x})}$ is the logistic function. Motion coherence levels are represented by $M$ (ranging from −1 to 1). The two tunable parameters are threshold (=inflection point) $t$, and the width (or sensitivity) $w$. We fitted the percentage right over coherence curves of single fish and the group average to the threshold-and-width-based psychometric curve using scipy's curve_fit function with default parameters (initial guess: $t = 0$, $w = 0.4$).

## Additive network model

Our additive network model takes four stimulus time series as input: motion_left, motion_right, luminance_left, and luminance_right. All input variables are between 0 and 1 and multiplied by the clutch-specific sensitivity factors to take into account some of the variability we observe across clutches (Supplementary Fig. 2a). To simulate leaky integration of the input signals, we designed four 30-second-long

exponential decay kernels, with time constants $\tau_m$, $\tau_{ll}$, $\tau_{lc}$, and $\tau_{mf}$: $K_m(t)$, $K_{ll}(t)$, $K_{lc}(t)$, and $K_{mf}(t)$:

$$K(t) = \frac{1}{\tau} * e^{-\frac{(t-15)}{\tau}} \text{ if } t > 15 \text{ else } 0 \tag{2.1}$$

$$M_L(t) = K_m(t) * \text{motion}_{\text{left}}(t) \tag{2.2}$$

$$M_R(t) = K_m(t) * \text{motion}_{\text{right}}(t) \tag{2.3}$$

$$LL_L(t) = K_{ll}(t) * \text{luminance}_{\text{left}}(t) \tag{2.4}$$

$$LL_R(t) = K_{ll}(t) * \text{luminance}_{\text{right}}(t) \tag{2.5}$$

$$LC_L(t) = K_{lc}(t) * \text{luminance}_{\text{left}}(t) \tag{2.6}$$

$$LC_R(t) = K_{lc}(t) * \text{luminance}_{\text{right}}(t) \tag{2.7}$$

Next, we detected changes from dark to bright (luminance increase) and vice versa (luminance decrease):

$$LI_L(t) = \max(\text{luminance}_{\text{left}}(t) - LC_L(t), 0) \tag{3.1}$$

$$LI_R(t) = \max\left(\text{luminance}_{\text{right}}(t) - LC_R(t), 0\right) \tag{3.2}$$

$$LD_L(t) = \max(LC_L(t) - \text{luminance}_{\text{left}}(t), 0) \tag{3.3}$$

$$LD_R(t) = \max\left(LC_R(t) - \text{luminance}_{\text{right}}(t), 0\right) \tag{3.4}$$

And we combined the luminance increase and decrease detectors to obtain luminance change detectors:

$$LC_L(t) = LI_L(t) + LD_L(t) \tag{4.1}$$

$$LC_R(t) = LI_R(t) + LD_R(t) \tag{4.2}$$

Then we took a weighted sum of the integrated motion, attractive luminance level and repulsive luminance change signals:

$$\text{multifeature}_{\text{left}}(t) = w_m M_L(t) + w_{ll} LL_L(t) + w_{lc} LC_R(t) \tag{5.1}$$

$$\text{multifeature}_{\text{right}}(t) = w_m M_R(t) + w_{ll} LL_R(t) + w_{lc} LC_L(t) \tag{5.2}$$

Next, we convolved the multifeature signals with $K_{mf}(t)$:

$$MF_L(t) = K_{mf}(t) * \text{multifeature}_{\text{left}}(t) \tag{6.1}$$

$$MF_R(t) = K_{mf}(t) * \text{multifeature}_{\text{right}}(t) \tag{6.2}$$

We found the ratio of left-swims:

$$\text{swims}_{\text{left}}(t) = \frac{1}{2}\left(\frac{MF_L(t) - MF_R(t)}{MF_L(t) + MF_R(t)} + 1\right) \tag{7}$$

Finally, we applied a 2-second rolling average window over this ratio, to match the bout analysis. To silence either of the three processing pathways in silico, we set either $w_m$, $w_{ll}$, or $w_{lc}$ to 0. Note that

this also removed the respective $\tau_m$, $\tau_{ll}$, or $\tau_{lc}$ from the free parameter list.

## Model fitting

We fitted the seven free parameters ($w_m$, $w_{ll}$, $w_{lc}$, $\tau_m$, $\tau_{ll}$, $\tau_{lc}$, and $\tau_{mf}$) by first splitting the data 50/50 in training and validation groups. Either we put all unifeature trials (motion OR luminance) in the training group and all combinations (motion AND luminance) in the validation group, or we put half the experiments across all trial types in the training group and the other half in the validation group.

During each of the 25 training rounds, we further split the training group into training and test sets. Two-thirds of the fish were part of the training set, and one-third of the fish were part of the test set. We then picked five times 20 random combinations of each trial kind and experiment type. For each of the five times, we set the boundaries and randomly picked the initial guess of the seven fitting parameters ($\tau_x$ in [0.1, 100], $w_x$ in [0.1, 25]). Using scipy's curve_fit() function, we fitted the additive network model to the training data set and obtained an MSE score by comparing the fitted model to the test data set. We stored the best out of five parameter sets for validation, meaning that we ended up with 25 parameter estimations. Testing these 25 parameter sets on the validation group gave us 25 validation MSE scores. We used these MSE scores to show whether unifeature training of the model was sufficient to explain motion/luminance combination trials. Finally, we used the median of each fitted parameter for visualization of the model node activities in Fig. 3.

## Two-photon imaging

We screened *Tg(elavl3:H2B-GCaMP8s)* and *Tg(alpha-tub:c3pa-GFP; elavl3:H2B-GCaMP6s)* larvae for GCaMP expression. At least 1 h prior to imaging, we embedded each larva within a 6 cm diameter Petri dish in low melting agarose ~2% (Ultra Pure Low Melting Point Agarose, 16520-100, Invitrogen) in E3 fish water.

After embedding, we transferred the fish under one of our custom-built two-photon microscopes. We controlled the microscopes through custom-written Python 3.12-based software (PyZebraPhysiology). In short, our microscopes consist of a shared tunable DeepSee MaiTai Ti:Sapphire laser (SpectraPhysics) operated at 950 nm to image GCaMP and tuned to 760 nm to photoactivate c3pa-GFP. We regulated the laser power with a combination of a lambda-half plate (Thorlabs, AHWP05M-980) and a Glan-Thomson prism (Thorlabs, GL5-A) to 12–15 mW at sample for functional imaging, and 5–7 mW at sample for c3pa-GFP activation. We scanned with a set of x/y galvanometers (Cambridge Technology). One microscope had a 20x NA 1.0 (XLUMPLFLN Olympus) objective, the other a 25x NA 1.1 (CFI75 Apo 25XC W 1300, Nikon) objective. We collected emitted light through GaAsP (green channel) and Alkali (red channel) photomultipliers (Hamamatsu) and amplified the signals with a current preamplifier (TIA60, Thorlabs). We presented visual stimuli through a mirror by a P300 Neo Pico Projector (AAXA Technologies) onto diffusive paper glued to the bottom of the experimental platform (8 cm diameter).

We scanned the functional imaging planes at ~1 Hz and 800 × 800 pixels at a resolution of 0.20 to 0.65 µm per pixel. The vertical distance between planes ranged from 2 to 12 µm. We imaged each plane for one hour and showed a random combination of nine different 1-minute stimuli, meaning that, on average, each stimulus was presented 6 to 7 times. We imaged 1 to 40 planes per fish. For the experiment to test the temporal integration properties of luminance integrators (Supplementary Fig. 3b,c), we imaged the optic tectum. We used a vertical distance between planes ranging from 12 to 15 µm. We imaged each plane for 45 min and showed a random combination of five different 1-min stimuli, meaning that, on average, each stimulus was presented nine times. We imaged 5 to 6 planes per fish. After each imaging session, we collected one overview stack (~50 planes, 800 ×800 pixels at a resolution of 0.65 ×0.65 ×2 µm) to allow anatomical registration to the

ZBRAIN and/or mapzebrain atlas[44,82]. All fluorescent images are displayed using linear colormaps.

## Imaging preprocessing

We preprocessed the raw imaging data using a custom-written Python 3.12-based preprocessing script. In brief, we performed CaImAn piecewise rigid motion-correction[83] followed by Cellpose automatic segmentation[84]. For the functional experiments and initial photoactivation experiments, the following parameters were used: model_type='cyto', diameter=12, cellpose_flow_threshold=0.95. For the rest of the photoactivation experiments, the following parameters were used: model_type='cyto3', cellpose_flow_threshold=0.4, cellpose_prob_threshold=0. We then used scipy.interp1d for stimulus temporal alignment (discretization=0.5 s), followed by a two-step ANTs anatomical registration to the ZBRAIN/mapzebrain atlases[44,82,85] (step 1=overview to atlas mapping, step 2 = functional plane to overview mapping). We used the following ANTs mapping parameters: 'interpolation_method': 'linear', 'use-histogram-matching': 0, 'matching_metric': 'MSE', 'rigid': {"t": "Rigid[0.1]", "m": "MI[$1,$2,1,32,Regular,0.25]", "c": "[1000x500x250×300,1e-8,10]", "s": "3x2x1×0", "f": "8x4x2×1"}, 'affine': {"t": "Affine[0.1]", "m": "MI[$1,$2,1,32,Regular,0.25]", "c": "[200x200x200×100,1e-8,10]", "s": "3x2x1×0", "f": "8x4x2×1"}, 'SyN': {"t": "SyN[0.1,6,0]", "m": "CC[$1,$2,1,2]", "c": "[200x200x200×100,1e-7,10]", "s": "4x3x2×1", "f": "12x8x4×2"}.

We normalized fluorescence $\frac{dF}{F_0} = \frac{F - F_0}{F_0}$, with $F_0$ being the average fluorescence during the pre-stimulus period (0 to 10 s).

## Imaging analysis with linear regression

We used the model predictions of the six nodes as regressors after convolving them with a GCaMP response kernel, an exponential decay with a time constant of 2.4 s, and unity-based normalization. We then used scipy's linregress function with default parameters to find the correlation coefficient between each regressor and the unity-based normalized average trace of each neuron. For the integrator nodes (motion, luminance, and multifeature), we used a threshold of 0.85, for the detector nodes (increase, decrease, and change) we used a threshold of 0.65.

## Imaging analysis with logical statements

To select which neurons fitted with the model predictions, we first summarized the response of each neuron into five values: A) Average pre-stimulus dF/F$_0$ (0 to 10 s), B) Average initial stimulus dF/F$_0$ (12.5 to 17.5 s), C) Average stimulus dF/F$_0$ (20 to 40 s), D) Average initial post-stimulus (42.5 to 47.5 s), and E) Average post-stimulus (50 to 60 s). We then used a set of logical statements to select our neurons of interest. M=Motion stimulus, L=Lateral luminance cue.

Motion integrators:

$$M_{\text{ipsi}}L_{\text{any}}C > T_{\text{activity}} \tag{8.1}$$

$$M_{\text{ipsi}}L_{\text{any}}C > M_{\text{ipsi}}L_{\text{any}}[A, B, E] \tag{8.2}$$

$$M_{[\text{off, contra}]}L_{[\text{ipsi, contra}]}C < T_{\text{minimum}} + M_{[\text{off, contra}]}L_{[\text{ipsi, contra}]}[A, E] \tag{8.3}$$

$$M_{[\text{off, contra}]}L_{[\text{ipsi, contra}]}D < T_{\text{minimum}} + M_{[\text{off, contra}]}L_{[\text{ipsi, contra}]}E \tag{8.4}$$

Multifeature integrators:

$$M_{\text{ipsi}}L_{\text{any}}C > T_{\text{activity}} \tag{9.1}$$

$$M_{\text{ipsi}}L_{\text{any}}C > M_{\text{ipsi}}L_{\text{any}}[A, E] \tag{9.2}$$

$$M_{[\text{off, contra}]}L_{\text{contra}}[B, D] > M_{[\text{off, contra}]}L_{\text{contra}}C \tag{9.3}$$

$$M_{\text{off}}L_{\text{ipsi}}C > M_{\text{off}}L_{\text{ipsi}}[A, E] \tag{9.4}$$

$$M_{\text{off}}L_{\text{contra}}B > M_{\text{off}}L_{\text{contra}}A \tag{9.5}$$

$$M_{\text{off}}L_{\text{contra}}D > M_{\text{off}}L_{\text{contra}}E \tag{9.6}$$

Luminance integrators:

$$M_{\text{any}}L_{\text{ipsi}}C > T_{\text{activity}} \tag{10.1}$$

$$M_{\text{any}}L_{\text{ipsi}}C > M_{\text{any}}L_{\text{ipsi}}[A, B, E] \tag{10.2}$$

$$M_{[\text{ipsi, contra}]}L_{\text{off}}C < T_{\text{minimum}} + M_{[\text{ipsi, contra}]}L_{\text{off}}[A, E] \tag{10.3}$$

$$M_{\text{any}}L_{\text{contra}}C < T_{\text{below}}M_{\text{any}}L_{\text{contra}}[A, E] \tag{10.4}$$

Luminance change detectors:

$$M_{\text{any}}L_{\text{contra}}B > T_{\text{activity}} \tag{11.1}$$

$$M_{\text{any}}L_{\text{contra}}[B, D] > M_{\text{any}}L_{\text{contra}}[A, C, E] \tag{11.2}$$

$$M_{\text{any}}L_{\text{ipsi}}[B, D] > M_{\text{any}}L_{\text{ipsi}}[A, C] \tag{11.3}$$

$$M_{\text{off}}L_{\text{contra}}B > T_{\text{peaksLC}}M_{\text{off}}L_{\text{ipsi}}B \tag{11.4}$$

Luminance increase detectors:

$$M_{\text{any}}L_{\text{contra}}B > T_{\text{activity}} \tag{12.1}$$

$$M_{\text{any}}L_{\text{contra}}B > M_{\text{any}}L_{\text{contra}}[A, C, D, E] \tag{12.2}$$

$$M_{\text{any}}L_{\text{ipsi}}D > M_{\text{any}}L_{\text{ipsi}}C \tag{12.3}$$

$$M_{\text{any}}L_{\text{ipsi}}B < T_{\text{minimum}} + M_{\text{any}}L_{\text{ipsi}}A \tag{12.4}$$

$$M_{\text{off}}L_{\text{contra}}B > T_{\text{peaks}}M_{\text{off}}L_{\text{ipsi}}D \tag{12.5}$$

$$M_{\text{off}}L_{\text{contra}}D < T_{\text{minimum}} + M_{\text{off}}L_{\text{contra}}A \tag{12.6}$$

Luminance decrease detectors:

$$M_{\text{any}}L_{\text{contra}}D > T_{\text{activity}} \tag{13.1}$$

$$M_{\text{any}}L_{\text{contra}}D > M_{\text{any}}L_{\text{contra}}[A, B, C, E] \tag{13.2}$$

$$M_{\text{any}}L_{\text{ipsi}}B > M_{\text{any}}L_{\text{ipsi}}A \tag{13.3}$$

$$M_{\text{any}}L_{\text{ipsi}}D < T_{\text{minimum}} + M_{\text{any}}L_{\text{ipsi}}A \tag{13.4}$$

$$M_{\text{off}}L_{\text{contra}}D > T_{\text{peaks}}M_{\text{off}}L_{\text{ipsi}}B \tag{13.5}$$

$$M_{off}L_{contra}B < T_{minimum} + M_{off}L_{contra}A \qquad (13.6)$$

WTA motion integrators:

$$M_{ipsi}L_{[off, ipsi]}C > T_{activity} \qquad (14.1)$$

$$M_{ipsi}L_{[off, ipsi]}C > M_{ipsi}L_{[off, ipsi]}[A, B, E] \qquad (14.2)$$

$$M_{[off, contra]}L_{[ipsi, contra]}C < T_{minimum} + M_{[off, contra]}L_{[ipsi, contra]}[A, E] \qquad (14.3)$$

$$M_{ipsi}L_{contra}C < T_{minimum} + M_{ipsi}L_{contra}[A, E] \qquad (14.4)$$

$$M_{[off, contra]}L_{[ipsi, contra]}D < T_{minimum} + M_{[off, contra]}L_{[ipsi, contra]}E \qquad (14.5)$$

WTA luminance integrators:

$$M_{[off, ipsi]}L_{ipsi}C > T_{activity} \qquad (15.1)$$

$$M_{[off, ipsi]}L_{ipsi}C > M_{[off, ipsi]}L_{ipsi}[A, B, E] \qquad (15.2)$$

$$M_{[ipsi, contra]}L_{off}C < T_{minimum} + M_{[ipsi, contra]}L_{off}[A, E] \qquad (15.3)$$

$$M_{ipsi}L_{contra}C < T_{minimum} + M_{ipsi}L_{contra}[A, E] \qquad (15.4)$$

$$M_{contra}L_{ipsi}C < T_{minimum} + M_{contra}L_{ipsi}[A, E] \qquad (15.5)$$

$$M_{any}L_{contra}C < T_{below}M_{any}L_{contra}[A, E] \qquad (15.6)$$

We then selected the neurons using thresholds: $T_{activity} = 0.2$, $T_{minimum} = 0.1$, $T_{peaksLC} = 1.25$, $T_{peaks} = 1.5$, and $T_{below} = 0.9$. Although there is an unavoidable bit of arbitrariness to the thresholds, we based the numbers on the overall noise levels throughout the data. The standard deviation of all averaged responses during the control stimulus (0% coherence, no lateral luminance cue) was 0.091. This means that our activity threshold results in selecting neurons whose activity is at least more than 2 standard deviations above baseline, whereas for example, the off-responses need to remain below 1 STD above baseline ($T_{minimum}$).

## Synthetic data simulation

To create synthetic data for our signal-to-noise tests (Supplementary Fig. 4e–h), we took the model node activity of the luminance change detectors and the luminance integrators. We normalized signals and added additional random spikes according to a Poisson distribution. We then convolved this activity trace with a GCaMP response kernel as described before under 'Imaging analysis linear regression'. Finally, we added Gaussian noise with a scale from 0.3 to 2.0. For each fitting round, we created 100 simulated leftward luminance change detectors, 100 simulated rightward luminance change detectors, 100 simulated leftward luminance integrators, and 100 simulated rightward luminance integrators. We then performed, for each noise-level, five times the logical statements based-classification as well as the linear regression based correlation analysis and calculated the percentage of correctly labeled luminance change detectors and luminance integrators.

## Exponential function fitting

To find the temporal integration properties of the luminance integrators (Supplementary Fig. 3b,c), we fitted an exponential function to

our data:

$$f(t) = a\left(1 - e^{-\frac{t}{\tau}}\right) + b \qquad (16)$$

We fitted this function to the median functional activity of all luminance integrators per fish using scipy's curve_fit() function with default parameters. We used $a = 1$, $\tau = 1$, $b = 0$ as initial guesses. We bounded $a$ by 0 and 100, $\tau$ by 0 and 60, and $b$ by −10 and 10.

## Neural manifold analysis

We performed our neural manifold analysis across all neurons that had seen at least 3 trials per stimulus. Due to the random stimulus order, there were a few planes where this was not the case, we excluded the neurons on those planes (~30000 neurons out of ~140000 were excluded).

We normalized all raw fluorescent traces and stacked stimuli, ordered as in Fig. 3b. We performed principal component analysis (PCA) using sklearn.decomposition's PCA function on either all neurons, or the neurons from a specific major brain region (forebrain, midbrain, hindbrain). We visualized the first three principal components (PCs) labeled by stimulus type (only-motion, only-luminance, congruent, and conflicting) or by stimulus direction (right and left).

Next, we calculated the Euclidean distance in 3-dimensional PCA space between left and right trials of the same type over stimulus time, as well as the control distance between trials of the same type and same direction.

## KDE binary mask generation and mapzebrain atlas-based anatomical analysis

To create the kernel density estimation (KDE) binary masks, we took the neurons per functional class as selected by our logical statements. We took the centroid locations and used the scipy's stats.gaussian_kde functionality with default parameters. We created a downsampled grid in ZBRAIN[82] coordinate space (10x downsampled for computational time reasons). We evaluated the density of this downsampled grid by putting it through the computed KDE and upsampled 10x to get to the original ZBRAIN coordinates and resolution. Finally, we binarized our mask by setting every pixel within either optic tectum or superior dorsal medulla oblongata stripe 1-3 in the left hemisphere with a density > $2^7$ to True. Next, we mapped the ZBRAIN coordinate space to mapzebrain coordinate space using a custom generated bridge transform (available upon request). Because of the high anatomical overlap, we took the union of the motion and multifeature masks, as well as the union of the four luminance related masks. We then searched the mapzebrain atlas for neurons that had their soma in either of the two masks.

For each anterior hindbrain neuron, we then checked whether it crossed the hemisphere ($x > 311$ in ZBRAIN units), whether it projected anterior ($y < 500$ ZBRAIN units), posterior ($y > 780$ ZBRAIN units), ventral ($z < 30$ ZBRAIN units), dorsal ($z > 90$ ZBRAIN units), and/or lateral ($x > 400$ ZBRAIN units). These thresholds were selected with the aim to separate projection patterns without having to manually evaluate each single neuron. We defined anterior_ipsi neurons to project anterior, but not dorsal nor ventral nor cross to the other hemisphere. Anterior_contra neurons project anterior, to the other hemisphere and lateral. Local_ipsi neurons do not project anterior, posterior, dorsal or cross the hemisphere. Local_contra neurons cross the hemisphere, but do not project anterior, posterior or dorsal.

For each tectal neuron, we checked whether it projected entirely within the tectum, whether it crossed the hemisphere ($x > 311$ in ZBRAIN units), projected anterior ($y < 400$ ZBRAIN units), posterior ($y > 780$ ZBRAIN units), ventral ($z < 50$ ZBRAIN units), and/or dorsal ($z > 120$ ZBRAIN units). We define front_pathway neurons as projecting anterior, outside the tectum, but not posterior nor crossing the hemisphere. Lateral_pathway neurons project ventral outside the

tectum, but not anterior nor posterior. Front_crossing neurons project anterior, outside the tectum, but not dorsal nor posterior. Posterior_pathway neurons project posterior outside the tectum.

In Supplementary Fig. 8b we plotted the local_ipsi anterior hindbrain neurons with the lateral_pathway tectal neurons; the anterior_ipsi anterior hindbrain neurons with the front_pathway tectal neurons; the anterior_contra anterior hindbrain neurons with the lateral_pathway tectal neurons; and the local_ipsi anterior hindbrain neurons with the lateral_cross_neurons.

## Photoactivations and morphological analyses

We performed functionally targeted photoactivations using a double-transgenic *Tg(alpha-tub:c3pa-GFP; elavl3:H2B-GCaMP6s)* line. As we have done previously[45], we outcrossed this line to *Tg(alpha-tub:c3pa-GFP)*, generating a high likelihood of offspring to be homozygous for *alpha-tub:c3pa-GFP* but heterozygous for *H2B-GCaMP6s*. Compared to other strategies using neuronal promoters and a single-construct design (FuGIMA)[35], our approach helped to strongly increase c3pa-GFP expression strength, while keeping GCaMP background fluorescence at lower levels. We embedded the fish as described above at least 2 h prior to the experiment. First, we checked for expression of c3pa-GFP by photoactivating an interneuron in the anterior right hemisphere of the tectal neuropil. These neurons project locally, so their photoactivation did not interfere with our later tracings of neurons in the left hemisphere.

We then functionally imaged 1 to 4 planes of interest for 9 minutes each, showing 3 repetitions of motion-left-luminance-off, motion-right-luminance-right, motion-off-luminance-left stimuli. These three stimuli allowed us to distinguish between all predicted cell types. We imaged at 950 nm, 8–10 mW, at a rate of ~1 Hz, with a pixel resolution of 0.21 ×0.21 μm. We manually selected our cells of interest using FIJIs[86] plot-z-axis-profile-live tool or using our custom-made Python 3.12-based software.

We photoactivated the selected neuron by performing 1 to 4 rounds of 20 ×200 ms-ON-100 ms-OFF pulses at 760 nm with 6–8 mW, focused on the center of the selected cell. Of these photoactivations, we obtained a success rate of 62% successfully targeted individual neurons (Supplementary Fig. 8d). Note that this rate is calculated after the selection of c3pa-GFP positive fish.

To prevent motion artifacts while imaging the photoactivated neuron, we anesthetized the fish with 300 μL of 0.015% MS-222 solution (Sigma-Aldrich, E10521-50G) directly after the photoactivation. While waiting for the MS-222 to diffuse through the agarose, we imaged a close-up stack of the photoactivated neuron (950 nm, 7–8 mW, 800 ×800 pixels, 30 to 40 planes, 0.042–0.084 ×0.042–0.084 × 0.5 μm resolution). This high resolution imaging allowed us to verify whether we truly hit a single neuron and none of its neighbors. About 20 min after the photoactivation, we imaged a broad volume around the neuron of interest (950 nm, 8–12 mW, 800 ×800 pixels, 60 to 100 planes, 0.65 ×0.65 ×2 μm resolution, 90 s average per plane). We initially traced a single neuron per fish. To increase efficiency, we later identified up to 8 additional neurons per fish, which we photoactivated and scanned after acquiring the broad volume of the single neuron.

We traced the photoactivated neurons in FIJI using the SNT tool[87], resulting in SWC files for each traced neuron. We registered these SWC files to the ZBRAIN/mapzebrain[44,82] atlas and extracted the brain regions (mapzebrain database downloaded on 19-06-2025 from the official website https://mapzebrain.org) through which each neuron passed.

## HCR-FISH

We performed functional experiments as described above, using the same 9 stimuli (at least 3 trials per stimuli), across at least 3 planes. For each fish, we covered the same region of the brain. The two-photon excitation wavelength was 950 nm, pixel lateral resolution of 0.3 μm, and frame rate was 1.3 Hz. We also recorded the volume using higher axial resolution (2 μm) to help registration with the post-in situ volume.

Larvae were fixed in ice-cold 4% paraformaldehyde (PFA) immediately after functional experiments, dehydrated in methanol, rehydrated, hybridized overnight at 37 °C with DNA probes, and fluorophore-conjugated hairpins were then added and amplified via hybridization chain reaction (HCR) according to standard procedures[88,89]. Probe sets and reagents were obtained from Molecular Instruments. We combined B1-*gad2*, B1-*gad1a*, B1-*gad1b*, B2-*slct17a6a* (vglut2b), B2-*slc17a6b* (vglut2a) probes, and used B1-546 and B2-405 as hairpins.

We then used the same two-photon system again to image HCR-FISH-labeled volumes. To detect the 546 nm red fluorophore, we imaged at an excitation wavelength of 1010 nm, simultaneously using two PMTs: one to detect green fluorescence from H2B-GCaMP8s and the other to detect red fluorescence from the HCR RNA-FISH probe. For detection of the 405 nm blue fluorophore, we imaged each z-plane at two excitation wavelengths. First, we used 950 nm excitation to image H2B-GCaMP8s with the first PMT. We then switched to 800 nm excitation to detect the blue fluorophore using the second PMT. In all configurations, laser power was 12–20 mW and planes were imaged for 30 to 120 s. Volumes were recorded at pixel lateral resolution of 0.3 microns, and axial resolution of 2 μm.

## HCR-FISH analysis

In situ imaging stacks were mapped to functional volume stacks using ANTs[85] with these parameters: 'rigid': {"use": True, "t": "Rigid[0.1]","m": "MI[$1,$2,1,32,Regular,0.25]", "c": "[1000x500x250×300,1e-8,10]", "s": "3x2x1×0","f": "8x4x2×1"}, 'affine': {"use": True, "t": "Affine[0.1]", "m": "MI[$1,$2,1,32,Regular,0.25]", "c": "[1000x500x250×100,1e-8,10]", "s": "3x2x1×0", "f": "8x4x2×1"}, 'SyN': {"use": True,"t": "SyN[0.1,6,0]", "m": "CC[$1,$2,1,2]", "c": "[200x200x200×100,1e-7,10]", "s": "4x3x2×1","f": "12x8x4×2"}.

Because GABAergic and glutamatergic neurons can be located adjacent to each other and alignment between the functional imaging volume and the post-fixation in situ-labeled volume can be imperfect, we thought that automatic assignment of neurotransmitter identity may be error-prone. To ensure accurate classification, we developed a custom-written GUI to manually inspect the alignment and assess *gad* and *vglut* expression for each neuron individually (Supplementary Fig. 6c). Only neurons that could clearly be identified as GABAergic or glutamatergic with high certainty were included in the analysis. Neurons were excluded if alignment was uncertain, if too few surrounding features were available to assess registration, if the labeling was too noisy, or if *gad* and *vglut* signals overlapped too strongly to allow clear discrimination. This approach led us to keep 315 out of the 2240 model-relevant neurons identified in these experiments for which neurotransmitter assignment confidence was very high.

## Statistics

Figure 1: Two-sided t-test for related samples to compare MSE between models ADD, WTA, MOT, and LUMI, we used Bonferroni correction for multiple tests. Cohen's d effect sizes; ADD vs. WTA: −1.17, ADD vs. MOT: −1.96, ADD vs. LUMI: −1.80. Two-sided t-test for related samples to compare relative interbout-interval, we used Bonferroni correction for multiple tests. Cohen's d effect sizes; M vs. L: −1.57, M = L vs M ≠ L: −0.84. Two-sided t-test for related samples to compare following-swims distributions, we used Bonferroni correction for multiple tests. Cohen's d effect size: M vs. L: −0.035, M vs M = L: −1.07, L vs M = L: −1.08. Figure 2: two-sided t-test for independent samples to compare the MSE distributions, we used Bonferroni correction for multiple tests. Cohen's d effect size: multifeature vs. unifeature full model: 0.39,

full model vs. silenced luminance change pathway: −2.58, full model vs. silenced luminance level pathway: −1.09, full model vs. silenced motion pathway: −5.60. Figure 4: two-sided t-test for independent samples between the excitatory and inhibitory ratio and 0.5 obtained no significant results. Supplementary Fig. 2: two-sided t-test for independent samples to compare the MSE distributions. Cohen's d effect size: 0.36. Supplementary Fig. 3: paired t-test between weak and medium luminance contrast, as well as between medium and strong contrasts with alternative 'greater'. We used Bonferroni correction for multiple tests. Cohen's d effect size: weak vs. medium: 2.16, medium vs. strong: 0.29. Supplementary Fig. 5: two-sided t-test for independent samples between mean distances during stimulus time for congruent and conflicting trials. Cohen's d effect size: 2.36 full brain, 2.74 rhombencephalon.

### Declaration of generative AI and AI-assisted technologies in the writing process

During the preparation of this manuscript, the authors used ChatGPT (GPT-4o) in order to refine readability and scientific tone. The authors then carefully reviewed and edited content. They take full responsibility for the content of the published article. All illustrations and schematics in the manuscript have been generated by the authors without generative AI, using the vector design program Affinity Designer 2.

### Lead contact

Requests for further information and resources should be directed to and will be fulfilled by Armin Bahl; armin.bahl@uni-konstanz.de

### Reporting summary

Further information on research design is available in the Nature Portfolio Reporting Summary linked to this article.

## Data availability

The behavior and preprocessed imaging data reported in this study are publicly available in KonData with the identifier https://doi.org/10.48606/tus7hs1zv77phbtv. The preprocessed imaging data contains the raw fluorescent traces for each neuron, as well as the averaged imaging stack. The raw imaging data itself is too large to upload to KonData (~ 3TB). This data will be provided upon request (armin.bahl@uni-konstanz.de) without restrictions. We will respond to requests within 10 working days, and shared data will remain available as long as requested.

## Materials availability

This study did not generate new unique reagents. The fish line generated in this study has been deposited at the ZFIN database (ID: ZDB-ALT-260121-5). The fish line will be made available upon request.

## Code availability

Analysis and model code have been archived and are publicly available in KonData with the identifier https://doi.org/10.48606/6jhypme0jeu49raa.

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

## Acknowledgements

We thank the members of the Bahl lab and of the other neurobiology groups in Konstanz, as well as Joseph Donovan and Iain Couzin, for fruitful discussions. We thank James Foster for advice on the psychometric analysis. We thank Ulrike Bonitz and our colleagues at the animal facility and the university workshop for their support. We thank Daniel Hummel, Fiona Klusmann, Ashrit Mangalwedhekar, Katrin Vogt, Joseph Donovan, and Margherita Zaupa for proofreading and feedback on this manuscript. This work was funded by the Emmy Noether Program (BA 5923/1-1), an ERC Starting Grant (101075541 – CollectiveDecisions), and the Deutsche Forschungsgemeinschaft (DFG, German Research Foundation) under Germany's Excellence Strategy (EXC 2117 – 422037984). In addition, the Zukunftskolleg Konstanz supported A. B. and M. Q. C. The International Max Planck Research School for Quantitative Behavior, Ecology and Evolution (IMPRS-QBEE) provided bridge funding for M. Q. C and K. Slangewal. K. Slangewal was also supported by a Boehringer Ingelheim Fonds graduate fellowship. A. B. and F. K. were supported via the National Institutes of Health U19 Program (U19NS104653).

## Author contributions

Conceptualization: K. Slangewal, M. Q. C., A. B.; Data curation: K. Slangewal, M. Q. C., F. K., S. A.; Formal analysis: K. Slangewal; Funding acquisition: A. B., K. Slangewal; Investigation: K. Slangewal, F. K., S. A., H. N.; Methodology: K. Slangewal, M. Q. C., F. K., S. A., H. N.; Project administration: K. Slangewal, A. B.; Resources: A. B.; Software and data curation: K. Slangewal, M. Q. C., F. K., S. A., A. B.; Supervision: A. B.; Validation: K. Slangewal, M. Q. C., A. B.; Visualization: K. Slangewal; Fish line generation (*elavl3:H2B-GCaMP8s*): H. B., K. Slanchev; Commenting and feedback on manuscript draft: S. A., M. Q. C., F. K., H. N., H. B. Writing manuscript, review, and editing: K. Slangewal, A. B.

## Funding

## Competing interests

The authors declare no competing interests.
