## [Transparent Peer Review file · Nature Communications]

Visuomotor decision-making through multifeature convergence in the larval zebrafish hindbrain

Corresponding Author: Professor Armin Bahl

Version 0:

Reviewer comments:

Reviewer #1

(Remarks to the Author)

In this study from the Bahl lab, the authors investigated how zebrafish larvae integrate multiple visual features (motion coherence, luminance level, and changes in luminance) while making decisions that guide swimming behavior. They first presented a clever set of experiments to carefully dissect visual feature integration in freely swimming zebrafish larvae. Based on the results of these behavioral experiments, they developed a model that mimics the animals' decision-making process. Subsequently, they used two-photon microscopy to identify neurons across the zebrafish brain that correspond to different nodes of the model. They discovered several such neurons with diverse properties in the midbrain and hindbrain circuitry. The combined results revealed that zebrafish employ an additive strategy, rather than a winner-takes-all approach, when processing these visual cues. The authors identified three parallel sensory pathways (motion, luminance level, and luminance change) that converge in the anterior hindbrain, which acts as a multi-feature integration hub. Interestingly, an earlier study showed that neurons in this anterior hindbrain region are related to the head direction system, further highlighting the importance of this area. One of the most exciting aspects of this study is the experimental design that enabled photo-labeling of functionally identified groups of neurons. This allowed the authors to determine the morphologies of functionally defined neurons. By combining open neural morphology data with their own functionally labeled anatomical data, the authors refined their working model, which is presented at the end of the paper and illustrates multi-feature integration of visual stimuli.

I believe this is an exciting study, paving the way for new approaches to studying brain networks. I am particularly enthusiastic about the authors' photo-activation approach, which enabled them to identify the morphologies of functionally defined neurons, a non-trivial task. The study has great potential, and I have a few comments and suggestions that I hope will help improve it:

Comments:

1. The biggest strength of the manuscript, the photo-labeling of functionally defined neurons, is not presented in sufficient detail to assess the quality and clarity of the method. For example, it is unclear how effective this labeling is in targeting the intended neurons. What is the success rate? How confident can we be that the labeled neuron is indeed the targeted one? As it stands, the presentation lacks sufficient clarity to evaluate potential pitfalls and the overall quality of this method.
2. Similarly, the quality of HCR matching to individual functionally imaged neurons is difficult to assess. Would it not be better to complement these HCR results with dual-channel imaging of calcium signals and neuronal identity using relevant vGlut and Gad1 transgenic lines?
3. The statement "Based on our previous structure-to-function analyses of motion processing neurons in the anterior hindbrain, we model these connections as likely inhibitory" seems to rely on a limited number of data points, expanding this data could be important. Is there no variability in the anatomy of functionally identified neurons? Are they truly so similar?
4. The anatomical descriptions are very zebrafish-centric and difficult to relate to general concepts of the vertebrate brain. Could the authors expand the discussion to relate their findings to what is known about vertebrate hindbrain circuits?
5. The authors state: "Furthermore, we conclude that motion and luminance cues jointly control swim initiation, leading to delayed responses in conflicting conditions." I could not find the specific data or results that support this conclusion. Could you please clarify?
6. The paragraph starting at line 211 is difficult to read and lacks sufficient figure citations.
7. The authors' results highlight the anterior hindbrain, a brain region recently shown to contain the head direction system in zebrafish. It would be important for the authors to discuss how their findings align with this earlier work.
8. The authors discuss visual responses in the habenula and cite a paper from the Julin Du lab, but they seem to have

missed the earlier paper by Dreosti et al. (2014), which first showed such visual responses. Please consider including this reference.

9. The y-axis in Figure 11 is not well defined.

10. Figure legends need clearer explanations, and some color codes are difficult to interpret.

11. The statements “We further find that the computation of lateral luminance change detection happens locally in the optic tectum” and “First, an excitatory connection of motion integrators onto multifeatured neurons without clear compartmentalization. Second, an inhibitory ipsilateral connection from lateral luminance level integrator cells onto multifeature neurons” are difficult to interpret. What specific datasets support these claims?

12. Similarly, the claims about dendritic computations on line 524 are not substantiated with data. I recommend toning down these statements.

(Remarks on code availability)

Reviewer #2

(Remarks to the Author)

Visuomotor decision-making through multifeature convergence in the larval zebrafish hindbrain

This study investigates the neural basis of multifeature visual integration in larval zebrafish. The authors combine precise behavioural assays with computational modeling to provide convincing evidence that zebrafish employ an additive algorithm to integrate three visual cues during sensorimotor decisions - coherent motion, steady-state luminance levels, and transient luminance changes. Using brain-wide two-photon calcium imaging, they map these computations to distinct neural populations, demonstrating motion and multifeature integration in the anterior hindbrain and luminance processing in the optic tectum. Complementary analyses of neurotransmitter identity (via HCR-FISH) and single-neuron morphology (via pa-GFP photoactivation) support a circuit model featuring parallel pathways that converge additively. The paper provides a biologically plausible framework linking behaviour to neural implementation, with broad implications for sensory integration across vertebrates. The work extends single-feature studies (e.g., motion in Bahl & Engert, *Nat. Neurosci.* 2020 [ref. 17]; luminance in Wolf et al., *Nat. Commun.* 2017 [ref. 22]) to multifeature contexts. Overall, I think that this is a very interesting paper that will have impact for future work across animal models used to investigate the neural basis of behaviour.

Significance: This advances our understanding of how vertebrate brains resolve sensory conflicts via parallel processing, with the anterior hindbrain emerging as a versatile integration hub. It addresses a core question in decision neuroscience—additive vs. competitive algorithms—while providing a blueprint for linking behaviour to circuits in compact brains. The multi-level analysis of this problem highlights the advantages of the larval zebrafish as a model. I can't think of a study using mice that similarly integrates behaviour, algorithmic modelling, brain-wide recording of activity (!) and pathway mapping. An excellent piece of work.

Minor concerns and suggestions

The manuscript has been prepared to a very high standard and documents the work in detail. The strongest parts of the manuscript are the behavioural analysis, the algorithmic modelling, and the survey and analysis of neural responses. The latter sections that attempt to link neural morphology/transmitter type to the circuit model were less satisfying overall, in large part because they lacked any experimental tests (e.g. by manipulating activity). But I don't think it would be reasonable to ask for this to be added to this manuscript (although it might warrant a mention in the Discussion).

The manuscript is already very strong, but a few clarifications would polish it further:

1. Figure 2 and model details: The luminance change pathway's high weight but modest silencing effect is well-explained, but quantifying its transient contribution (e.g., via integrated activity over stimulus edges) could strengthen the intuition. Consider adding a supplemental simulation showing pathway contributions across time.

2. Neural classification: Logical statements are effective, but the thresholds (e.g., $\Delta F/F_0 > 0.1$) feel somewhat arbitrary—briefly justify them in Methods (e.g., based on noise floors) or via a ROC curve in supplements. The winner-takes-all cells (Extended Data Fig. S3D) are intriguing; a sentence speculating on their recruitment (e.g., under time pressure) could tie to the discussion's flexibility theme.

3. General: There is a tendency to repetition in the manuscript, especially in the Discussion. Readability could be improved by cutting some of this and making the Discussion more focused. For instance, it feels like there is some overreach in cross-species comparisons. The claim of “conserved principles” ignores algorithmic differences—mammalian multisensory integration often involves divisive normalization (Ohshiro et al., *Nat. Neurosci.* 2011 [ref. 57]), not just addition. The authors' mention of sensory processing disorders feels cursory to this reviewer, although this is a matter of taste.

Leon Lagnado

(Remarks on code availability)

Reviewer #3

(Remarks to the Author)

The paper "Visuomotor decision-making through multifeatured convergence in the larval zebrafish hindbrain" by Slangewal et al., describes behavioral and functional responses across the zebrafish brain to reorientation-inducing visual stimuli. Given previous literature on motion integration and phototaxis, the data obtained in this manuscript aren't necessarily surprising. However, the manuscript combines responses to luminance differences and lateral motion in a clever way. The central claim of the paper is that luminance differences and motion direction are processed in parallel pathways. Given the proposed architecture of visual processing parallel processing of features is not surprising. However, I am not sure if the presented data supports the term "parallel pathways" well enough, see below. While experiments and analysis are well done overall, I have strong concerns with some of the interpretations which need to be addressed before publication.

-Major concerns-

1 The concept of "parallel pathways"

A major finding according to the authors is the integration of information from two parallel "pathways" a motion pathway and a luminance pathway. Given visual system architecture and responses, parallel processing is not surprising. However, the term "parallel pathways" suggests that information is kept separate across the circuits until it hits the "multifeature integrators." The experiments conducted by the authors are not sufficient to draw this conclusion. The experiments demonstrate that at the level of behavior, there is no multiplicative interaction while the imaging experiments suggest a separation of the pathways, except the "multifeature integrators." However, the experiments cannot show that these multifeature integrators are in fact at the end of pathways rather than in the middle, etc. In the end, a well-supported conclusion is that the information adds linearly and that there are neural populations that separately process the information as well as neural populations that might integrate the information (more on that below). I would therefore suggest the authors speak of "parallel processing" and "linear addition" instead of "parallel pathways."

2 Luminance "integrators" and "multifeature integrators"

Previous work has demonstrated beautifully that neurons in the anterior hindbrain integrate lateral motion cues. A hallmark of this integration is that motion coherence affects the speed of integration but not the reachable level. In fact, this is critical information to conclude that a neural population integrates information. Simply showing that activity rises slowly is insufficient, especially with calcium imaging. It is therefore misleading to name the yellow population (top right) in Figure 3F "Luminance integrators" despite their match to model predictions for one luminance level. The authors will need to show that the activity slope within these neurons depends on luminance contrast while the level reached does not. Absent that, the neurons should be given a name that reflects that the neurons show sustained responses to luminance differences. This is the only conclusion that can be drawn given the current data.

In a similar vein, I don't think that the blue population in the same figure (bottom right) can be confidently labeled "Multifeature integrators." There are clear differences between the responses and the predictions which highlights an overall problem: How is it decided that the match "is good enough" to warrant the name given by the model?

3 Orthogonal processing of motion and luminance

Given the current analysis, this statement is misleading. To properly characterize the presumed "manifold", the dimensionality of the data needs to be considered. Given the fraction of explained variance, it has to be assumed that this dimensionality is larger than 30 as 30 components only explain a little more than 50% of the variance. Statements about orthogonality will need to consider the full dimensionality. What the authors find in Figure S4 is therefore not a feature of the data but a feature of the analysis. This section does not add information while being used to draw a misleading conclusion - the section title suggests the existence of a "low-dimensional space" and "orthogonal processing." I also do not believe that this information is important for the overall work - the authors already show a separation of encoding on the individual neuron level. Considering the model structure, this seems to be the more important result.

4 Prior work

Since a lot of the features discovered by the authors are predicted by prior work in the visual system, there should be a broader discussion on properties of visual responses and how these relate to the findings. The discussion instead seems to unnecessarily focus on highlighting perceived advances of the authors' approach which may or may not be warranted.

-Minor points-

"Winner take-all responses": The definition of the authors is not sufficient to label these types of neurons as "winner take-all." The neurons the authors describe with this name are just linear adders (with negative sign), while "winner take-all" cells should respond to the more important feature, either in a constant or stochastic manner.

Neuron identification by logical statements: This is a clever idea. However, the fact that fewer neurons are identified compared with regression is not enough to conclude that the approach is better or worse. Maybe I'm missing something and the statement needs to be clarified or additional analysis is needed such as decreases in false-positive-rates on shuffled data with this approach compared to regression.

Anatomy: I really like the PA-GFP approach, however, this section is confusing as currently written and it did not become clear to me how the findings related to strengthening or refuting parts of the model proposed by the authors. I also believe that this section could be tightened considerably by removing the initial single-cell MapZBrain analysis which doesn't seem

to address any important points that aren't better made based on the PA-GFP conversion.

Anatomy 2: The PA-GFP conversions (at least as described) suggest that no neurons cross the mid-hindbrain boundary. How does this chime with the proposed model and the claim that luminance neurons in the tectum feed information to hindbrain neurons?

(Remarks on code availability)

Reviewer #4

(Remarks to the Author)

We co-reviewed this manuscript with one of the reviewers who provided the listed reports. This is part of the Nature Communications initiative to facilitate training in peer review and to provide appropriate recognition for Early Career Researchers who co-review manuscripts.

In this study, the authors use an extensive combination of computational and experimental tools to dissect the pathways underlying sensory integration of luminance and motion. The behavioral and neurological bases of each modality in isolation have been characterized in previous studies. The novelty here lies in combining them and uncovering an additive behavioral strategy that is also reflected in brain-wide imaging data. Despite rigorous experiments, the authors do not arrive at a precise mapping for their computational model's units within the zebrafish brain, nor do they determine how these units are interconnected. Although this may be perceived as a weakness of the study, the authors convincingly apply a diverse set of cutting-edge techniques, generating multiple circuit-level hypotheses, and demonstrating that simple sensory computations can be distributed across many brain regions and cell types, even in a small vertebrate brain. The innovative workflow of this study provides compelling evidence for the distributed nature of neural circuits, from sensory to higher-order networks. This work will be impactful for future efforts to map specific computations at whole-brain scale in vertebrates. We see no major flaw that would compromise the validity of the study, and consider that the study is thorough and meets the standards for publication in Nature Communications. Some minor/moderate comments are provided below to improve readability, some aspects of the discussion, and the interpretation of key results.

Moderate comments:

Is the proposed model in Fig. 3E the simplest possible algorithm that performs these computations? Given that, after all these experiments, the authors do not arrive at a decisive mapping of the model to the brain, there may be other, potentially more sophisticated models with similar internal dynamics that are implemented by the brain. The possibility of other algorithmic implementations should be discussed.

In a similar idea, the activity patterns predicted by the model (Fig. 3F) are quite simple (constant activity during stimulus presentation, activation at stimulus onset and offset, etc). It is not surprising that there are neurons displaying these dynamics, regardless of whether the brain is actually implementing the model or not. Moreover, the presence of neural signals matching the expected model dynamics does not guarantee their causal involvement in the computations. Numerous studies in mice and primates have highlighted how task-related signals can be decoded from most brain regions (for a recent example: <https://www.nature.com/articles/s41586-025-09235-0>). Complementarily, lesion studies show that many of these regions are in fact unnecessary for task performance. This should be discussed as one of the main limitations of the study, which does not conclude with a clear one-to-one mapping between the model's units and the larval brain. The search for neural signatures in large-scale neural datasets comes with the ever-present risk of spurious correlations. The authors do a good job of reinforcing their functional imaging results with anatomy and grounded circuit hypotheses, therefore we are not concerned for the study's validity. However, this is an important discussion element that should be further developed.

Minor comments:

Some figure legends are quite long. For instance, Fig. 1A contains many methodological details (size of the arena, camera frame rate, etc) that are not required to interpret the figure panel. In contrast, Fig. 3A is an equivalent panel for the imaging setup, yet has a much shorter description.

There are a few narrative shortcuts. Some passages should be better contextualized to improve the accessibility of the manuscript, especially for non-zebrafish researchers. For instance, the expected behavioral response of zebrafish larvae to luminance and motion is not clearly stated. The authors mention that the stimuli "drive directional swimming", but in which specific direction (i.e. toward light) should be made explicit before presenting the first behavioral results. The ethological significance of these stimuli and behaviors should also be discussed briefly and early on in the manuscript.

Similarly, the authors build on previous studies by Bahl and colleagues, without explicitly stating what these studies demonstrated: that higher levels of motion coherence increase the likelihood of swimming in the direction of motion. Without this context, it may be initially unclear what the "psychometric curve" represents for the uninitiated reader when first introduced in Fig. 1E,F.

Regarding Fig. 1K, I'm not sure why shorter IBIs in congruent stimuli and longer IBS in conflicting stimuli provide evidence for convergent processing. A more mechanistic account of parallel vs convergent processing could be provided first to better interpret those results. In the PCA analysis section, the better separability of congruent stimuli is interpreted as a population-level mechanism for faster reaction times, which does make sense, as better separability may lead to better downstream

decoding and faster conversion to motor outputs. While these results are compelling and complement the main analysis, the authors should clarify further how this provides strong evidence for “parallel processing followed by convergent additive integration”.

In Fig. 3D, the neurons are compiled across animals, but the number of larvae/experiments is not indicated.

Could the uneven coverage of the brain in the calcium imaging experiments bias the anatomical mapping of the functional units and the interpretation of the circuit, especially if there is variability across animals? There is substantial variability in the E:I ratios in Fig. 4D. Could this variability stem from an uneven sampling due to the position of the imaging planes? Perhaps there could be a way, using atlas markers, to estimate how plane positioning may have biased the E:I estimates (though the feasibility of this might be a problem, given that atlas markers are mostly population averages that blur out single-cell information).

In Fig. 3F, the open circles are difficult to perceive when not zoomed in on the figure. Moreover, the fact that neurons matching a given model hemisphere are also mostly located within the same brain hemisphere is an interesting observation that is currently not (and should be) addressed in the main text.

“We obtained a relatively high number of principal components”: this should be rephrased, as PCA does not return an arbitrary number of components. A certain number of components is required to explain a certain fraction of variance.

“We performed hybridization chain reaction fluorescent in situ hybridization (HCR-FISH) to determine whether functionally identified neurons were (glutamatergic; vglut) or (GABAergic; gad).” The authors should state more clearly how these experiments are carried out in the main text. Larvae are first imaged in vivo, then fixed, then stained, and then re-imaged post hoc, before being computationally registered onto the calcium imaging experiments. This is a long pipeline, and although it is becoming a more common procedure, these experiments are impressive and should be elaborated a bit further in the main text.

“This balance is likely required for the implementation of the temporal integration properties within each computational unit.” Do the authors say this because the timescale of a computational unit greatly exceeds the synaptic/single-neuron timescale? If so, then it should be stated more explicitly.

On the same topic, the authors could speculate a bit further on why the E:I ratios are so balanced and why the implementation of the algorithm over distributed neuronal populations may be a desirable feature, beyond “shaping output dynamics” and “the implementation of specific temporal dynamics”. In principle, a handful of neurons could have implemented this circuit, yet this is not the case. One possible reason is that larger circuits are more robust to noise. Another is that they may allow for pruning without altering the network output too much (this may be especially relevant in a developing brain).

Typos: “winter-take-all”, “e1evl3”, ;)

(Remarks on code availability)

We have not run the Python code to reproduce the results, but inspected the files and the code is well-organized, with each figure and supplementary figure supported by its own script. However, the repository is minimally documented, and we recommend the authors improve its presentation and accessibility if the paper is to be accepted and published.

Reviewer #5

(Remarks to the Author)

(Remarks on code availability)

I have not run the Python code to reproduce the results, but I inspected the files and the code is well-organized, with each figure and supplementary figure supported by its own script. However, the repository is minimally documented, and I recommend the authors improve its presentation and accessibility if the paper is to be accepted and published.

Version 1:

Reviewer comments:

Reviewer #1

(Remarks to the Author)

The authors successfully addressed my comments. I also think that they have done substantial amount of work to address other reviewers comments successfully too. I recommend the publication of this manuscript at Nature Communications.

(Remarks on code availability)

I am not at the capacity to assess the code details.

Reviewer #2

(Remarks to the Author)

The manuscript has been significantly improved and I judge it to be of a very high standard and of high interest.

(Remarks on code availability)

Reviewer #3

(Remarks to the Author)

I thank the authors for engaging with my comments and focusing the conclusions of the paper on aspects that are fully supported by the data. I think that the paper overall presents a very nice dissection of stimulus features triggering visually guided behaviors, and that it should be published in its current form.

(Remarks on code availability)

Reviewer #4

(Remarks to the Author)

the authors have addressed each of our points through appropriate textual revisions and additional figures. We have no remaining concerns and consider the revised manuscript significantly improved.

(Remarks on code availability)

Reviewer #5

(Remarks to the Author)

(Remarks on code availability)

The GitHub has been updated and now has better documentation.

made.

Point-by-point response to reviewer comments

Reviewer comments highlighted in yellow. Response directly below each point.

REVIEWER COMMENTS

Reviewer #1 (Remarks to the Author):

In this study from the Bahl lab, the authors investigated how zebrafish larvae integrate multiple visual features (motion coherence, luminance level, and changes in luminance) while making decisions that guide swimming behavior. They first presented a clever set of experiments to carefully dissect visual feature integration in freely swimming zebrafish larvae. Based on the results of these behavioral experiments, they developed a model that mimics the animals' decision-making process. Subsequently, they used two-photon microscopy to identify neurons across the zebrafish brain that correspond to different nodes of the model. They discovered several such neurons with diverse properties in the midbrain and hindbrain circuitry. The combined results revealed that zebrafish employ an additive strategy, rather than a winner-takes-all approach, when processing these visual cues. The authors identified three parallel sensory pathways (motion, luminance level, and luminance change) that converge in the anterior hindbrain, which acts as a multi-feature integration hub. Interestingly, an earlier study showed that neurons in this anterior hindbrain region are related to the head direction system, further highlighting the importance of this area. One of the most exciting aspects of this study is the experimental design that enabled photo-labeling of functionally identified groups of neurons. This allowed the authors to determine the morphologies of functionally defined neurons. By combining open neural morphology data with their own functionally labeled anatomical data, the authors refined their working model, which is presented at the end of the paper and illustrates multi-feature integration of visual stimuli.

I believe this is an exciting study, paving the way for new approaches to studying brain networks. I am particularly enthusiastic about the authors' photo-activation approach, which enabled them to identify the morphologies of functionally defined neurons, a non-trivial task. The study has great potential, and I have a few comments and suggestions that I hope will help improve it:

Comments:

1. The biggest strength of the manuscript, the photo-labeling of functionally defined neurons, is not presented in sufficient detail to assess the quality and clarity of the method. For example, it is unclear how effective this labeling is in targeting the intended neurons. What is the success rate? How confident can we be that the labeled neuron is indeed the targeted one? As it stands, the presentation lacks sufficient clarity to evaluate potential pitfalls and the overall quality of this method.

Thank you for pointing out the strength of the functionally guided photoactivations. We now provide images of 8 example cells in **Supplementary Fig. 8c**. We also added a breakdown of our success rate (single target neuron successfully labeled = 62%) and reasons for failure (**Supplementary Fig. 8d**). We did this quantification manually by carefully comparing pre-

and post-photoactivations. Based on the alignment of neighboring cell bodies before and after photoactivation, we are highly confident that the labeled cell is truly the targeted one. To further verify the labeling of single neurons, we also generated close-up volume stacks around the labeled neurons (**Supplementary Movie 4**). It is clearly visible that after photoactivation, we only see a single neurite leaving the targeted cell body.

2. Similarly, the quality of HCR matching to individual functionally imaged neurons is difficult to assess. Would it not be better to complement these HCR results with dual-channel imaging of calcium signals and neuronal identity using relevant vGlut and Gad1 transgenic lines?

We agree with the referee that an assessment of the mapping quality would help the reader to appreciate the precision of our labeling strategy. We have developed a graphical user interface for an expert annotator to load ANTs-registered datasets. ANTs alone achieves good, but not perfect, matching, requiring this manual inspection step. Our approach further helps as the shapes of cells are largely variable, and HCR FISH distribution in the cytosol is better quantifiable through a manual analysis. We provide a new **Supplementary Fig. 6** to show 8 randomly chosen cells to compare the fixed to functional mapping, as well as the neurotransmitter identity labeling. The new figure also shows a screenshot of the GUI illustrating the rigor of our annotation process. This display shows that we can make neurotransmitter assignments with very high confidence to functionally characterized cells.

We agree that the dual-channel imaging using vGlut and Gad1 transgenic lines is a good idea, and it may simplify mapping between live- and fixed-imaging stacks. However, we only have *gad1b:dsRed* and *vglut2a:dsRed* lines (Satou et al. 2013) in our lab. This means, in combination with GCaMP imaging (green), we would only be able to assign excitatory or inhibitory labels to functionally identified cells per fish, but not both within the same experiment. We would require one of these transgenic lines expressing fluorophores of different colors (such as blue), which, to our knowledge, do not exist. In addition, with HCR FISH, we can label multiple variants of *gad* (*gad1a*, *gad1b*, *gad2*) and *vglut* (*vglut2a*, *vglut2b*), enhancing confidence about the excitatory/inhibitory labels.

3. The statement “Based on our previous structure-to-function analyses of motion processing neurons in the anterior hindbrain, we model these connections as likely inhibitory” seems to rely on a limited number of data points, expanding this data could be important. Is there no variability in the anatomy of functionally identified neurons? Are they truly so similar?

Thanks for pointing out that this needs further clarification. In this paper, we have performed the neurotransmitter labeling after functional imaging without morphological analysis (**Fig. 4**). The anatomical analyses (**Fig. 5**) were done after functional imaging, however, without HCR FISH. Therefore, we cannot fully constrain the neurotransmitter labels within specific anatomical projection groups, as these results come from different experiments and fish. However, in our recent paper (*Boulanger-Weill et al. 2025, BioRxiv*), we performed correlated light and electron microscopy in the anterior hindbrain in fish with inhibitory neurons fluorescently labeled. We could thus obtain information about the neurotransmitter identity as

a function of anatomical projection pattern at the same time. 26 out of 29 reconstructed motion-responsive cells with contralateral projections were Gad1b+. Based on the result of *Boulanger-Weill et al. 2025*, we modeled the projections in this current paper as likely inhibitory. We hope the referee agrees that the size of this dataset provides a sufficient number of data points for the statement. To better explain our reasoning, we have updated and refined the main text of the manuscript accordingly.

Our reconstructions also show that there is indeed cell-to-cell variability within functional types, for example, for the motion and multifeature integrators. For each cell type, we grouped those different anatomies into 'anterior', 'contralateral', and 'local', based on where the projections go (**Fig. 5d**). We also added the number of neurons in each of the subplots of **Fig. 5d** to the figure legend for clarification.

4. The anatomical descriptions are very zebrafish-centric and difficult to relate to general concepts of the vertebrate brain. Could the authors expand the discussion to relate their findings to what is known about vertebrate hindbrain circuits?

Thanks for pointing this out. We added further information comparing our work to other vertebrate species (see the paragraph starting at line 603). Please note that we have limited these additions to a minimum, as per the request of Reviewer #2 (point 3), not to overreach in cross-species comparisons, and not to over-interpret these analogies as direct homologies.

5. The authors state: "Furthermore, we conclude that motion and luminance cues jointly control swim initiation, leading to delayed responses in conflicting conditions." I could not find the specific data or results that support this conclusion. Could you please clarify?

Thank you for pointing out that this has not been clear enough. We based our conclusion on the different interbout intervals for congruent stimuli compared to conflicting stimuli (**Fig. 1k**). We better explain our reasoning in (see lines 183–187) in our updated manuscript.

6. The paragraph starting at line 211 is difficult to read and lacks sufficient figure citations.

We rewrote the paragraph to be clearer and added more relevant figure citations using more precise panel labeling.

7. The authors' results highlight the anterior hindbrain, a brain region recently shown to contain the head direction system in zebrafish. It would be important for the authors to discuss how their findings align with this earlier work.

We thank the referee for pointing out that the anterior hindbrain contains neurons related to a navigational head direction system. Our and others' findings indicate that this region also represents multi-feature stimuli. These results together align well with work in the *Drosophila* central complex, which integrates multiple sensory cues across modalities.

We are now discussing this important point in our updated manuscript (the paragraph starting at line 595).

8. The authors discuss visual responses in the habenula and cite a paper from the Julin Du lab, but they seem to have missed the earlier paper by Dreosti et al. (2014), which first showed such visual responses. Please consider including this reference.

We included the missing reference.

9. The y-axis in Figure 11 is not well defined.

We added the unit and updated the plot.

10. Figure legends need clearer explanations, and some color codes are difficult to interpret.

We adjusted our figure legends to be clearer and more concise (as also requested by Reviewers #4 and #5, point 3). We improved the color code in **Fig. 2** to reduce the number of shades of blue, making it easier to interpret the figure. If the referee agrees, we would like to keep the coloring style in the **Figs. 3–5**. They are color-blind friendly arrangements without added unnecessary complexity.

11. The statements “We further find that the computation of lateral luminance change detection happens locally in the optic tectum” and “First, an excitatory connection of motion integrators onto multifeatured neurons without clear compartmentalization. Second, an inhibitory ipsilateral connection from lateral luminance level integrator cells onto multifeatured neurons” are difficult to interpret. What specific datasets support these claims?

Thank you for pointing out that these sentences were not clear enough. They both describe how we refine our behaviorally constrained model. We rephrased this section and added references to the corresponding figure panels to better justify our argumentation. See paragraph starting at line 496.

12. Similarly, the claims about dendritic computations on line 524 are not substantiated with data. I recommend toning down these statements.

Thanks, and we agree. We further toned down the statement, now on line 519.

Reviewer #2 (Remarks to the Author):

Visuomotor decision-making through multifeature convergence in the larval zebrafish hindbrain

This study investigates the neural basis of multifeature visual integration in larval zebrafish.

The authors combine precise behavioural assays with computational modeling to provide convincing evidence that zebrafish employ an additive algorithm to integrate three visual cues during sensorimotor decisions - coherent motion, steady-state luminance levels, and transient luminance changes. Using brain-wide two-photon calcium imaging, they map these computations to distinct neural populations, demonstrating motion and multifeature integration in the anterior hindbrain and luminance processing in the optic tectum. Complementary analyses of neurotransmitter identity (via HCR-FISH) and single-neuron morphology (via pa-GFP photoactivation) support a circuit model featuring parallel pathways that converge additively. The paper provides a biologically plausible framework linking behaviour to neural implementation, with broad implications for sensory integration across vertebrates. The work extends single-feature studies (e.g., motion in Bahl & Engert, Nat. Neurosci. 2020 [ref. 17]; luminance in Wolf et al., Nat. Commun. 2017 [ref. 22]) to multifeature contexts. Overall, I think that this is a very interesting paper that will have impact for future work across animal models used to investigate the neural basis of behaviour.

Significance: This advances our understanding of how vertebrate brains resolve sensory conflicts via parallel processing, with the anterior hindbrain emerging as a versatile integration hub. It addresses a core question in decision neuroscience—additive vs. competitive algorithms—while providing a blueprint for linking behaviour to circuits in compact brains. The multi-level analysis of this problem highlights the advantages of the larval zebrafish as a model. I can't think of a study using mice that similarly integrates behaviour, algorithmic modelling, brain-wide recording of activity (!) and pathway mapping. An excellent piece of work.

Minor concerns and suggestions

The manuscript has been prepared to a very high standard and documents the work in detail. The strongest parts of the manuscript are the behavioural analysis, the algorithmic modelling, and the survey and analysis of neural responses. The latter sections that attempt to link neural morphology/transmitter type to the circuit model were less satisfying overall, in large part because they lacked any experimental tests (e.g. by manipulating activity). But I don't think it would be reasonable to ask for this to be added to this manuscript (although it might warrant a mention in the Discussion).

We thank the referee for this encouraging and positive overall assessment. We extended our discussion on future work with a focus on the need for causal manipulations, providing an example of what such manipulations may test for (line 646).

The manuscript is already very strong, but a few clarifications would polish it further:

1. Figure 2 and model details: The luminance change pathway's high weight but modest silencing effect is well-explained, but quantifying its transient contribution (e.g., via integrated activity over stimulus edges) could strengthen the intuition. Consider adding a supplemental simulation showing pathway contributions across time.

We appreciate your idea on how to strengthen the intuition of our result. We added a panel

to **Supplementary Fig. 2c**, which demonstrates the simulated pathway contribution across time.

2. Neural classification: Logical statements are effective, but the thresholds (e.g., $\Delta dF/F0 > 0.1$) feel somewhat arbitrary—briefly justify them in **Methods** (e.g., based on noise floors) or via a ROC curve in supplements. The winner-takes-all cells (Extended Data Fig. S3D) are intriguing; a sentence speculating on their recruitment (e.g., under time pressure) could tie to the discussion's flexibility theme.

We agree that thresholds add an unavoidable bit of arbitrariness to functional imaging studies. We added a justification based on the noise levels in our data to the **Methods** (lines 973–977), indicating that threshold values are more than 2 standard deviations above baseline noise floors. We also added a sentence to our discussion speculating on the advantage of recruiting winner-takes-all cells under increased time pressure (lines 622-625).

3. General: There is a tendency to repetition in the manuscript, especially in the Discussion. Readability could be improved by cutting some of this and making the Discussion more focused. For instance, it feels like there is some overreach in cross-species comparisons. The claim of “conserved principles” ignores algorithmic differences—mammalian multisensory integration often involves divisive normalization (Ohshiro et al., Nat. Neurosci. 2011 [ref. 57]), not just addition. The authors’ mention of sensory processing disorders feels cursory to this reviewer, although this is a matter of taste.

Leon Lagnado

Thanks for pointing this out. We made our discussion more focused and improved readability. Given the request of Reviewer #1 (point 4), we have refined the text where we compare our findings with work done in other species, while limiting overreach in cross-species comparisons (the paragraph starting at line 603). Following the referee's recommendation, we now also mention divisive normalization as an alternative integration strategy (line 608–610).

Reviewer #3 (Remarks to the Author):

The paper “Visuomotor decision-making through multifeatured convergence in the larval zebrafish hindbrain” by Slangewal et al., describes behavioral and functional responses across the zebrafish brain to reorientation-inducing visual stimuli. Given previous literature on motion integration and phototaxis, the data obtained in this manuscript aren’t necessarily surprising. However, the manuscript combines responses to luminance differences and lateral motion in a clever way. The central claim of the paper is that luminance differences and motion direction are processed in parallel pathways. Given the proposed architecture of visual processing parallel processing of features is not surprising. However, I am not sure if the presented data supports the term “parallel pathways” well enough, see below. While experiments and analysis are well done overall, I have strong concerns with some of the interpretations which need to be addressed before publication.

-Major concerns-

1 The concept of “parallel pathways”

A major finding according to the authors is the integration of information from two parallel

“pathways” a motion pathway and a luminance pathway. Given visual system architecture and responses, parallel processing is not surprising. However, the term “parallel pathways” suggests that information is kept separate across the circuits until it hits the “multifeature integrators.” The experiments conducted by the authors are not sufficient to draw this conclusion. The experiments demonstrate that at the level of behavior, there is no multiplicative interaction while the imaging experiments suggest a separation of the pathways, except the “multifeature integrators.” However, the experiments cannot show that these multifeature integrators are in fact at the end of pathways rather than in the middle, etc. In the end, a well-supported conclusion is that the information adds linearly and that there are neural populations that separately process the information as well as neural populations that might integrate the information (more on that below). I would therefore suggest the authors speak of “parallel processing” and “linear addition” instead of “parallel pathways.”

Thank you for pointing out that ‘parallel processing’ and ‘linear addition’ better reflect our findings. We have adjusted our wording throughout our updated manuscript, and now only use the suggested terms and specify pathways as ‘processing pathways’ or ‘computational pathways’.

2 Luminance “integrators” and “multifeature integrators”

Previous work has demonstrated beautifully that neurons in the anterior hindbrain integrate lateral motion cues. A hallmark of this integration is that motion coherence affects the speed of integration but not the reachable level. In fact, this is critical information to conclude that a neural population integrates information. Simply showing that activity rises slowly is insufficient, especially with calcium imaging. It is therefore misleading to name the yellow population (top right) in Figure 3F “Luminance integrators” despite their match to model predictions for one luminance level. The authors will need to show that the activity slope within these neurons depends on luminance contrast while the level reached does not. Absent that, the neurons should be given a name that reflects that the neurons show sustained responses to luminance differences. This is the only conclusion that can be drawn given the current data.

In a similar vein, I don’t think that the blue population in the same figure (bottom right) can be confidently labeled “Multifeature integrators.” There are clear differences between the responses and the predictions which highlights an overall problem: How is it decided that the match “is good enough” to warrant the name given by the model?

We thank the reviewer for helping us to clarify the naming of model nodes in our manuscript. Regarding the luminance integrators, we have performed a new imaging experiment, following the referee's advice. Our new data shows slower integration for a weaker lateral luminance cue, while reaching similar activity levels. Therefore – and we hope the referee agrees – we feel confident that these cells can indeed be labeled as luminance integrators. We added the results to **Supplementary Fig. 3b,c**.

Also, we appreciate the referee's comment regarding the multifeature integrators. Whether a match in imaging data is considered ‘good enough’ to represent the model node is determined by our logical statements (see also point 6). For the multifeature neurons, the statements ensure that each selected neuron responds to all three features: motion, luminance, and

changes in luminance, as well as the combinations of these features. Given these criteria, we are confident to use the label 'multifeature integrator'.

Notably, our logical statement approach does not precisely fit model traces to neural data over time, but rather focuses on key features of the response. Therefore, residual differences between model response dynamics and experimental data are expected.

The most obvious difference between the model and the data is the decreasing response below baseline of motion integrators and multifeature integrators in the left hemisphere when motion is presented to the right, and vice versa for the right hemisphere. A possible explanation is the existence of contralateral mutual inhibition of motion integrators and/or multifeature integrators. Our anatomical analyses (**Fig. 5**) identify contralateral projections between motion integrators across hemispheres as well as contralateral projections of multifeature integrators across hemispheres (**Fig. 5f**). These projections could explain the mismatch in dynamics in **Fig. 3f**. We describe these issues now in our updated manuscript (lines 521–527 and 616–619).

3 Orthogonal processing of motion and luminance

Given the current analysis, this statement is misleading. To properly characterize the presumed "manifold", the dimensionality of the data needs to be considered. Given the fraction of explained variance, it has to be assumed that this dimensionality is larger than 30 as 30 components only explain a little more than 50% of the variance. Statements about orthogonality will need to consider the full dimensionality. What the authors find in Figure S4 is therefore not a feature of the data but a feature of the analysis. This section does not add information while being used to draw a misleading conclusion - the section title suggests the existence of a "low-dimensional space" and "orthogonal processing." I also do not believe that this information is important for the overall work - the authors already show a separation of encoding on the individual neuron level. Considering the model structure, this seems to be the more important result.

We agree that using terms like orthogonal processing was not appropriately phrased, in particular given the dimensionality of our data. We have adjusted the title of this section and shortened the text to focus our findings on the separability of congruent vs. conflicting stimuli in low-dimensional neural space.

We think that including an analysis framework in reduced-dimensional neural space adds value to our manuscript. Such unsupervised methods are commonly used in neuroscience and allow one to explore the brain dynamics without a model-guided strategy. Indeed, our results on the separability of different stimulus classes nicely match our findings from the behavior: For congruent stimulus cases, we have found that behavioral response delays are shorter than for conflicting configurations (**Fig. 1K**), which is reflected by distinct neural dynamics in low-dimensional space (**Supplementary Fig. 5d**). For these reasons, we would kindly ask to keep this supplementary information, after the mentioned adjustments, if the referee agrees.

4 Prior work

Since a lot of the features discovered by the authors are predicted by prior work in the visual system, there should be a broader discussion on properties of visual responses and how these relate to the findings. The discussion instead seems to unnecessarily focus on highlighting perceived advances of the authors' approach which may or may not be warranted.

We more rigorously link our results to prior work in the visual system through an improved discussion section. We have also adjusted our text to tone down statements related to the conceptual and methodological advances of our study.

-Minor points-

"Winner take-all responses": The definition of the authors is not sufficient to label these types of neurons as "winner take-all." The neurons the authors describe with this name are just linear adders (with negative sign), while "winner take-all" cells should respond to the more important feature, either in a constant or stochastic manner.

Thank you for catching this mistake, and we apologize for our incorrect explanation. The winner-takes-all model – two types of neurons connected through mutual inhibition – (**Supplementary Fig. 4d**) behaves as the referee points out: When driven by similar input, one of them will stochastically take over, inhibiting the other one. We have now updated the text (lines 369–372).

Neuron identification by logical statements: This is a clever idea. However, the fact that fewer neurons are identified compared with regression is not enough to conclude that the approach is better or worse. Maybe I'm missing something and the statement needs to be clarified or additional analysis is needed such as decreases in false-positive-rates on shuffled data with this approach compared to regression.

Thank you for pointing out that our statement requires further validation. We have now performed additional new analyses to address this concern. Specifically, we used our model (**Fig. 3e**) to generate neural data for two functional cell types, one with transient dynamics (luminance change detectors) and one with constant dynamics (luminance integrators). We then added different amounts of Gaussian noise to those traces. We then checked to what degree our logical statements and linear regression analysis would label neurons correctly. Quantification of the percentage of correctly classified neurons showed that our logical statement strategy does indeed perform better than linear regression for both cell types, in particular when signal-to-noise levels get lower. We added this additional perspective to **Supplementary Fig. 4e–h** and now explain this approach in detail in the text (lines 354-366) and in the Methods under 'Synthetic data simulation'.

We also added the false-positive rate on shuffled data for linear regression (**Supplementary Fig. 4b**), which is similarly low to the false-positive rate for our logical-statement-based approach.

Anatomy: I really like the PA-GFP approach, however, this section is confusing as currently written and it did not become clear to me how the findings related to strengthening or refuting parts of the model proposed by the authors. I also believe that this section could be tightened considerably by removing the initial single-cell MapZBrain analysis which doesn't seem to address any important points that aren't better made based on the PA-GFP conversion.

We largely restructured and rewrote this section of the results to improve its clarity and highlight how our findings relate to the updates of the model. See text from line 496 onwards.

Thank you for providing a suggestion on how to tighten this section. We think the mapzebrain analysis provides important context to our study, considerably narrowing down the possible projection patterns and neural diversity from the tectum to the anterior hindbrain. Without such knowledge, it would be difficult to know the order of magnitude of neurons needed to be generated through the time-consuming function-guided PA-GFP experiments. Moreover, this public database contains an increasingly large number of reconstructed cells, which will further facilitate hypothesis-driven circuit neuroscience over the next years. Showing that the neuronal morphologies downloaded from the database match the morphologies identified through photoactivation validates both the public platform as well as our method. We shortened our text related to this part and moved mapzebrain-based anatomies to **Supplementary Fig. 8a,b**.

Anatomy 2: The PA-GFP conversions (at least as described) suggest that no neurons cross the mid-hindbrain boundary. How does this chime with the proposed model and the claim that luminance neurons in the tectum feed information to hindbrain neurons?

Indeed, the referee is right about that we did not find any mid-hindbrain crossings. The luminance neurons in the tectum (which is part of the midbrain) reach the multifeature neurons of the anterior hindbrain through two boundaries. The tectal neurons project ventrally and cross the midbrain-forebrain boundary. The multifeature neurons project ventrally and anteriorly and cross the hindbrain-forebrain boundary. Both neuron types then converge in the ventral forebrain. For an illustration of the brain region arrangement, please see **Fig. 3b**. We have improved the explanation of these structural features in our updated results section (lines 489–495).

Reviewer #4 (Remarks to the Author):

We co-reviewed this manuscript with one of the reviewers who provided the listed reports. This is part of the Nature Communications initiative to facilitate training in peer review and to provide appropriate recognition for Early Career Researchers who co-review manuscripts.

In this study, the authors use an extensive combination of computational and experimental tools to dissect the pathways underlying sensory integration of luminance and motion. The behavioral and neurological bases of each modality in isolation have been characterized in previous studies. The novelty here lies in combining them and uncovering an additive

behavioral strategy that is also reflected in brain-wide imaging data. Despite rigorous experiments, the authors do not arrive at a precise mapping for their computational model's units within the zebrafish brain, nor do they determine how these units are interconnected. Although this may be perceived as a weakness of the study, the authors convincingly apply a diverse set of cutting-edge techniques, generating multiple circuit-level hypotheses, and demonstrating that simple sensory computations can be distributed across many brain regions and cell types, even in a small vertebrate brain. The innovative workflow of this study provides compelling evidence for the distributed nature of neural circuits, from sensory to higher-order networks. This work will be impactful for future efforts to map specific computations at whole-brain scale in vertebrates. We see no major flaw that would compromise the validity of the study, and consider that the study is thorough and meets the standards for publication in Nature Communications. Some minor/moderate comments are provided below to improve readability, some aspects of the discussion, and the interpretation of key results.

Moderate comments:

Is the proposed model in Fig. 3E the simplest possible algorithm that performs these computations? Given that, after all these experiments, the authors do not arrive at a decisive mapping of the model to the brain, there may be other, potentially more sophisticated models with similar internal dynamics that are implemented by the brain. The possibility of other algorithmic implementations should be discussed.

From the behavior, we see that there are three fundamental components: Animals respond to motion cues, to steady-state lateral luminance cues, as well as to the changes in luminance. As such, we need three components in our model to capture key aspects of the behavior. To our knowledge, no simpler model can capture our results. To explicitly probe the contributions, we algorithmically silenced individual pathway components, identifying that each of them is important to describe aspects of the behavior (**Fig. 2h**).

We agree with the referee that more complex models than the proposed one are likely to be implemented by the brain. In more naturalistic settings, animals need to cope with other challenges, they flexibly adapt across development, and they socially interact. All these components would need to be iteratively explored and added to the model to increase its complexity. Furthermore, apparently even simple computations, such as taking the temporal derivative of a luminance signal, can involve sophisticated temporal processing, which is the reason why identified computational representations are more complex than simple unit-to-neuron mapping.

We think the minimalist algorithmic modelling is an important starting point for the dissection of neural circuits. Indeed, the model allowed us to develop precise hypotheses to search for computational nodes in the brain, which we then identified. Analyzing anatomical details of these neurons then enabled us to expand and refine our model (**Fig. 5f**). This shows that a behavioral model-based circuit dissection loop can iteratively lead to a better understanding of the functioning of neural circuits and should provide a key strategy in neuroscience.

We have addressed these important points in the discussion (paragraph starting at line 613).

In a similar idea, the activity patterns predicted by the model (Fig. 3F) are quite simple (constant activity during stimulus presentation, activation at stimulus onset and offset, etc). It is not surprising that there are neurons displaying these dynamics, regardless of whether the brain is actually implementing the model or not. Moreover, the presence of neural signals matching the expected model dynamics does not guarantee their causal involvement in the computations. Numerous studies in mice and primates have highlighted how task-related signals can be decoded from most brain regions (for a recent example: <https://www.nature.com/articles/s41586-025-09235-0>). Complementary, lesion studies show that many of these regions are in fact unnecessary for task performance. This should be discussed as one of the main limitations of the study, which does not conclude with a clear one-to-one mapping between the model's units and the larval brain. The search for neural signatures in large-scale neural datasets comes with the ever-present risk of spurious correlations. The authors do a good job of reinforcing their functional imaging results with anatomy and grounded circuit hypotheses, therefore we are not concerned for the study's validity. However, this is an important discussion element that should be further developed.

We agree that the presence of neural signals matching the expected model dynamics does not guarantee their causal involvement in behavior. We further highlighted the need for causal manipulations in future studies to complement this work (lines 645–651).

We also agree with the referee that searching for neural signatures across large-scale datasets bears the risk of false-positive correlations. We would like to point out that we have probed our analyses using control configurations where stimuli were replaced with 0 % coherence on a homogenous background (**Supplementary Fig. 4b,c**). Here, even though we only take data without directional stimuli, we do find a few cells matching our logical statements and functional regressors; however, these cells were much fewer in number without spatial arrangements. Therefore, we conclude that the majority of the identified cells using our visual stimuli are real representations of the explored computation. We strengthened this point in the respective result section of our updated manuscript (lines 361–366).

Minor comments:

Some figure legends are quite long. For instance, Fig. 1A contains many methodological details (size of the arena, camera frame rate, etc) that are not required to interpret the figure panel. In contrast, Fig. 3A is an equivalent panel for the imaging setup, yet has a much shorter description.

Thank you for your comment. We adjusted the figure legends (also based on the comment of Reviewer #1, point 10) and removed unnecessary methodological details.

There are a few narrative shortcuts. Some passages should be better contextualized to improve the accessibility of the manuscript, especially for non-zebrafish researchers. For instance, the expected behavioral response of zebrafish larvae to luminance and motion is not clearly stated. The authors mention that the stimuli “drive directional swimming”, but in

which specific direction (i.e. toward light) should be made explicit before presenting the first behavioral results. The ethological significance of these stimuli and behaviors should also be discussed briefly and early on in the manuscript.

Thanks for pointing this out and providing explicit examples. We added clarification for these behaviors and their ethological relevance to the introduction (lines 70–72) as well as to the results part (lines 147–151).

Similarly, the authors build on previous studies by Bahl and colleagues, without explicitly stating what these studies demonstrated: that higher levels of motion coherence increase the likelihood of swimming in the direction of motion. Without this context, it may be initially unclear what the “psychometric curve” represents for the uninitiated reader when first introduced in Fig. 1E,F.

Thank you, we added further context and explained expectations (lines 147–151).

Regarding Fig. 1K, I’m not sure why shorter IBIs in congruent stimuli and longer IBS in conflicting stimuli provide evidence for convergent processing. A more mechanistic account of parallel vs convergent processing could be provided first to better interpret those results. In the PCA analysis section, the better separability of congruent stimuli is interpreted as a population-level mechanism for faster reaction times, which does make sense, as better separability may lead to better downstream decoding and faster conversion to motor outputs. While these results are compelling and complement the main analysis, the authors should clarify further how this provides strong evidence for “parallel processing followed by convergent additive integration”.

We adjusted the text to clarify our reasoning regarding **Fig. 1K** (lines 183–187). For the comment on our PCA analysis, please see our response to reviewer #3, point 3. Following reviewer #3’s advice, we largely rewrote the section for the PCA-based analysis.

In Fig. 3D, the neurons are compiled across animals, but the number of larvae/experiments is not indicated.

We clarified in the figure legend of **Fig. 3** that the N=15 fish corresponds to the data across the entire figure.

Could the uneven coverage of the brain in the calcium imaging experiments bias the anatomical mapping of the functional units and the interpretation of the circuit, especially if there is variability across animals? There is substantial variability in the E:I ratios in Fig. 4D. Could this variability stem from an uneven sampling due to the position of the imaging planes? Perhaps there could be a way, using atlas markers, to estimate how plane positioning may have biased the E:I estimates (though the feasibility of this might be a problem, given that atlas markers are mostly population averages that blur out single-cell

information).

We thank the referee for bringing this to our attention and for the constructive feedback. We now better illustrate the brain-wide distribution of functionally classified cell types. To this end, we improved the text to focus the reader on **Supplementary Fig. 3a**, in which we display brain region and cell-number normalized ratios. This plot thus corrects for the uneven coverage of our imaging as well as brain-region size differences. We argue for chosen regions for anatomical mapping based on this plot as well as based on general literature references.

We are confident that the variability in the E:I ratios in **Fig. 4d** is not a result of uneven coverage bias. Because in this experiment, we ensured near-identical imaging conditions of exactly the same volume across animals. We added a sentence to the methods to clarify the imaging configuration in this experiment (lines 1083–1087).

In Fig. 3F, the open circles are difficult to perceive when not zoomed in on the figure. Moreover, the fact that neurons matching a given model hemisphere are also mostly located within the same brain hemisphere is an interesting observation that is currently not (and should be) addressed in the main text.

We adjusted the style of the open circles to make it easier to distinguish the open vs filled circles. We adjusted this in all brain maps in **Fig. 3f-g, Fig. 4d, Supplementary Fig 3b, Supplementary Fig. 4a-c, and Supplementary Fig. 7a**. We now also highlight the hemispheric symmetry (lines 342-344).

“We obtained a relatively high number of principal components”: this should be rephrased, as PCA does not return an arbitrary number of components. A certain number of components is required to explain a certain fraction of variance.

True, thank you for finding this mistake. We rephrased the sentence to ‘*We found that 30 principal components were necessary to explain 53% of the total variance*’.

“We performed hybridization chain reaction fluorescent in situ hybridization (HCR-FISH) to determine whether functionally identified neurons were (glutamatergic; vglut) or (GABAergic;gad).” The authors should state more clearly how these experiments are carried out in the main text. Larvae are first imaged in vivo, then fixed, then stained, and then re-imaged post hoc, before being computationally registered onto the calcium imaging experiments. This is a long pipeline, and although it is becoming a more common procedure, these experiments are impressive and should be elaborated a bit further in the main text.

Thank you for the recognition of our experimental efforts. We now better explain the pipeline in the main text (lines 414–419). Moreover, we added a new **Supplementary Fig. 6**, for the detailed experimental and computational registration procedures, as well as displaying the quality of mapping results.

“This balance is likely required for the implementation of the temporal integration properties within each computational unit.” Do the authors say this because the timescale of a computational unit greatly exceeds the synaptic/single-neuron timescale? If so, then it should be stated more explicitly.

Yes, exactly, we made this sentence more explicit (lines 530–531).

On the same topic, the authors could speculate a bit further on why the E:I ratios are so balanced and why the implementation of the algorithm over distributed neuronal populations may be a desirable feature, beyond “shaping output dynamics” and “the implementation of specific temporal dynamics”. In principle, a handful of neurons could have implemented this circuit, yet this is not the case. One possible reason is that larger circuits are more robust to noise. Another is that they may allow for pruning without altering the network output too much (this may be especially relevant in a developing brain).

Thank you for the suggestions on these important discussion points. Robustness, temporal integration, and developmental stability are indeed key features of neural networks, and we have now improved our discussion of these aspects. (lines 631–636).

Typos: “winter-take-all”, “elevel3”, ;)

Thank you for catching those! ;-)

Reviewer #4 (Remarks on code availability):

We have not run the Python code to reproduce the results, but inspected the files and the code is well-organized, with each figure and supplementary figure supported by its own script. However, the repository is minimally documented, and we recommend the authors improve its presentation and accessibility if the paper is to be accepted and published.

We thank the referees for acknowledging our efforts. We have further improved our code documentation, which should largely improve readability.

Reviewer #5 (Remarks to the Author):

Reviewer #5 (Remarks on code availability):

I have not run the Python code to reproduce the results, but I inspected the files and the code is well-organized, with each figure and supplementary figure supported by its own script. However, the repository is minimally documented, and I recommend the authors improve its presentation and accessibility if the paper is to be accepted and published.

We thank the referees for acknowledging our efforts. We have further improved our code documentation, which should largely improve readability.